

# The MATS Satellite Mission − Gravity Waves Studies by Mesospheric Airglow/Aerosol Tomography and Spectroscopy

Jörg Gumbel[1], Linda Megner[1], Ole Martin Christensen[1,2], Seunghyuk Chang[3], Joachim Dillner[1], Terese Ekebrand[4], Gabriel Giono[5], Arvid Hammar[4,6], Jonas Hedin[1], Nickolay Ivchenko[5,7], Bodil Karlsson[1], Mikael Kruse[4], Anqi Li[2], Steven McCallion[4], Donal P. Murtagh[2], Georgi Olentšenko[5], Soojong Pak[8], Woojin Park[8], Jordan Rouse[4], Jacek Stegman[1], Georg Witt[1]

[1]Department of Meteorology (MISU), Stockholm University, Stockholm, Sweden
[2]Earth and Space Sciences, Chalmers University of Technology, Göteborg, Sweden
[3]Center for Integrated Smart Sensors, KAIST Dogok Campus, Seoul, Republic of Korea
[4]Omnisys Instruments AB, August Barks gata 6B, Västra Frölunda, Sweden
[5]School of Electrical Engineering, Royal Institute of Technology (KTH), Stockholm, Sweden
[6]Department of Microtechnology and Nanoscience, Chalmers University of Technology, Göteborg, Sweden
[7]South African National Space Agency, Hermanus 7200, South Africa
[8]School of Space Research, Kyung Hee University, Yongin-si, Republic of Korea

*Correspondence to*: Jörg Gumbel (gumbel@misu.su.se)

**Abstract.** Global three-dimensional data are a key to understanding gravity wave interactions in the mesosphere and lower thermosphere. MATS (Mesospheric Airglow/Aerosol Tomography and Spectroscopy) is a new Swedish satellite mission that addresses this need. It applies space-borne limb imaging in combination with tomographic and spectroscopic analysis to obtain gravity wave data on relevant spatial scales. Primary measurement targets are $O_2$ Atmospheric Band dayglow and nightglow in the near infrared, and sunlight scattered from noctilucent clouds in the ultraviolet. While tomography provides horizontally and vertically resolved data, spectroscopy allows analysis in terms of mesospheric temperature, composition, and cloud properties. Based on these dynamical tracers, MATS will produce a climatology on wave spectra during a 2-year mission. Major scientific objectives concern a characterization of gravity waves and their interactions in the mesosphere and lower thermosphere, as well as their relationship to dynamical conditions in the lower and upper atmosphere. MATS is currently being prepared for launch in 2019. This paper provides an overview over scientific goals, measurement concepts, instruments, and analysis ideas.

# 1 Introduction

## 1.1 Gravity waves in the mesosphere and lower thermosphere

Atmospheric gravity waves are small scale buoyancy waves that can transport momentum and energy over large distances in the atmosphere. Primary sources are disturbances in the troposphere such as flow over topography, convective systems or jets. Conservation of energy makes the amplitude of the gravity waves grow as they propagate upward towards less dense altitudes. As the waves dissipate, they deposit their momentum and energy onto the background atmosphere. This in turn affects the



atmosphere over a wide range of scales, from the local generation of turbulence to the forcing of large scale circulation (Fritts and Alexander, 2003; Alexander et al., 2010). This dynamical forcing is most prominent in the mesosphere and lower thermosphere (MLT), at altitudes of typically 50-130 km. Here a large fraction of upward propagating gravity waves reach their maximum amplitudes and break. The resulting dynamical forcing causes a global-scale circulation in the mesosphere

with strong upwelling at the summer pole and downwelling at the winter pole (Lindzen, 1981; Holton, 1982). Adiabatic cooling and heating connected to this circulation causes thermal conditions in the mesosphere to deviate far from radiative equilibrium. As one consequence, the polar summer mesosphere is turned into the coldest place on Earth, with temperatures reaching well below 130 K despite permanently sunlit conditions. This makes the region the home of the highest clouds on Earth, noctilucent clouds (NLC) (Thomas, 1991; Karlsson and Shepherd, 2018).

The role of gravity waves is further complicated as they interact with the background flow on their way through the middle atmosphere. This leads to an altitude-dependent filtering of the gravity wave spectrum by the wind field (e.g., Fritts and Alexander, 2003), including the wind patterns connected to planetary waves or tidal waves. The gravity wave spectrum arriving at higher altitudes thus carries an imprint of the dynamics at lower altitudes. This leads to a number of interesting teleconnections that can link conditions in widely separated regions of the atmosphere. Examples are the control of the summer

mesosphere by lower atmospheric conditions in terms of inter- and intra-hemispheric coupling (Gumbel and Karlsson, 2011; Körnich and Becker, 2010). Interactions between gravity waves and the mean flow can also give rise to a generation of secondary waves in the mesosphere. Gravity waves can generate planetary waves, either directly through zonally non-uniform dissipation (Holton, 1984), or indirectly through induced baroclinic instability in the vicinity of jets (Plumb, 1983; Sato and Nomoto, 2015). The breakdown of gravity waves can also generate secondary gravity waves, propagating both upward and

downward. This happens through localized momentum and energy fluxes, which in turn create strong body forces and imbalances (Vadas et al., 2003; Fritts et al., 2006; Becker and Vadas, 2018).

While the basic nature of the wave-driven circulation of the middle atmosphere is today understood, important mechanisms and interactions remain to be quantified. Decisive quantities for the forcing of the mean flow are the directional momentum flux and the altitude distribution of wave dissipation. Today, many general circulation models can explicitly simulate gravity

waves with longer horizontal and vertical wavelengths, while shorter sub-grid waves need to be parameterized e.g. in terms of the "wave drag" that they exert on the  mean flow (Alexander et al., 2010; Geller et al, 2013). Rather than accounting for the detailed underlying physics, these wave parameterizations are often used as a means of tuning the model to ensure realistic output e.g. in terms of middle atmospheric wind fields or temperature fields. Recently, important steps have been taken towards completely gravity-wave-resolving general circulation models (Watanabe et al., 2015; Becker and Vadas, 2018).

Complementary to these developments, ray-tracing models are important tools for case studies of wave events and comparisons to specific observational datasets (e.g., Marks and Eckerman, 1995; Kalisch et al., 2014). A goal of ongoing model developments is to explicitly describe the entire chain from the lower atmospheric source region, via the lower and middle atmospheric wave filtering, to the wave effects in the MLT. Only wave-resolved simulations can be expected to describe the





physics of e.g. intermittent wave interactions, energy cascading in the Lorenz energy cycle, or turbulence generation (Becker, 2012).

Observational data are critical for supporting such model developments, in particular in the MLT where gravity wave effects are most evident. Unfortunately, we are today lacking global observations of wave spectra arriving in the MLT, and

even more so of the contribution of different parts of the wave spectra to momentum transfer. Such observations are desirable not only to constrain model results directly in the MLT. Rather, MLT data can serve as a benchmark for testing wave implementations throughout the lower and middle atmosphere: general circulation models need to correctly describe the chain of wave processes at all altitudes in order to correctly reproduce resulting gravity wave properties observed in the MLT. In addition to providing a relevant database of gravity wave spectra, MLT studies are also needed that allow investigations of the

three-dimensional structure of wave propagation. An important example is the refraction of gravity waves in the vicinity of the mesospheric jet (Sato et al., 2009, Preusse et al., 2009, Ern et al, 2011). Other three-dimensional propagation effects concern the interaction of gravity waves with the polar vortex (McLandress et al., 2012; de Wit et al., 2014; Wright et al., 2017) or the suggested refraction of gravity waves during sudden stratospheric warmings (Thurairajah et al., 2014, Ern et al., 2016). Also concerning mesospheric ice formation and NLC, explanations are lacking for a number of dynamical features like so-called

ice fronts or ice voids (Megner et al., 2018).

While the above descriptions have focused on wave interactions from the lower atmosphere to the MLT, the importance of gravity waves extends well beyond these altitudes. The MLT can be regarded as a transition region where many fundamental changes occur in atmospheric properties. Examples are the transition from well-mixed, turbulent conditions to molecular diffusion, a transition to non-local thermodynamic equilibrium with a substantially increased lifetime of excited species, a

transition to an extreme-UV radiative environment, or the transition to increasing importance of ionospheric processes. Various "layered phenomena" in the MLT can be regarded as manifestations of these transitions. Prominent phenomena include dayglow and nightglow, layers of metal, dust or ice, as well as various plasma processes – all demonstrating strong links to both below and above. Despite this fact, the altitude around 100 km has long been regarded as a dividing line between different research communities, separating the middle and upper atmosphere. This view has changed in recent decades, and has today

been replaced by a strong interest in "whole atmosphere" model approaches that emphasize the connecting rather than the dividing role of the MLT (e.g., Roble, 2000; Marsh et al., 2007; Akmaev, 2011). Wave processes play a central role in this respect, and in bridging the atmospheric communities.

In addition to comprehensive whole atmosphere modeling efforts, there has been growing observational evidence of thermospheric and ionospheric responses to wave processes in the lower and middle atmosphere. The dynamical morphology

of thermosphere and ionosphere has been shown to be strongly connected to tidal waves (e.g., Anderson, 1981; Oberheide et al., 2009), but also to planetary waves (e.g., Chen, 1992; Forbes and Leveroni, 1992) and gravity waves (e.g., Röttger, 1977; Park et al., 2014; Forbes et al., 2016; Trinh et al., 2018). A basic open question concerns the relative importance of primary and secondary waves in propagating from the middle atmosphere to the thermosphere and ionosphere (Becker and Vadas, 2017). In the altitude range 100-300 km, gravity waves have been shown to create temperature variations of 50 K and density



variations of 10-25% over spatial scales of tens to hundreds of kilometres (Vadas and Liu, 2013). As the gravity waves interact with the background flow before reaching these altitudes, fingerprints of middle atmospheric circulation systems have been revealed well into the thermosphere and ionosphere (Siskind et al., 2012). A prominent example is dynamic coupling suggested to occur during sudden stratospheric warmings (Funke et al., 2010; Chau et al., 2011). Akmaev (2011) estimates that more

than half of the regular daily and seasonal variability in the thermosphere and ionosphere is forced from below. Akmaev further concludes that the availability of global data on MLT dynamics and variability is a limiting factor for future scientific progress, thus contrasting the MLT to the "data-rich" lower atmosphere and upper thermosphere.

In summary, there is substantial need for global observations of gravity waves in the mesosphere and lower thermosphere. These datasets are needed to support and verify ongoing developments of general circulation models, concerning both gravity

wave parameterizations and gravity-wave-resolved implementations. These datasets should provide: (1) information about horizontal (and vertical) wave spectra in order to identify the dominant scales that govern interactions with the mean flow, larger scale waves, and possibly secondary waves, (2) information about directional momentum flux as decisive quantity for these interactions, and (3) three-dimensional wave information in order to address detailed propagation and refraction effects in the vicinity relevant dynamical structures.

**1.2 Satellite measurements of gravity waves**

Where are we concerning such global observations of gravity waves in the MLT? Various satellite missions have provided data on MLT structures that have been analysed in terms of wave activity on various scales. Most of these apply limb-viewing geometries in a number of spectral ranges. On the TIMED satellite, infrared limb measurements by the SABER instrument provide species and temperature distributions that allow for retrievals of gravity waves and planetary waves (Krebsbach and

Preusse, 2007; Forbes et al, 2009; Preusse et al., 2009). The TIDI instrument provides MLT gravity wave data in terms of airglow Doppler wind measurements (Liu et al., 2009). On the ENVISAT satellite, infrared limb emission measurements by the MIPAS instrument cover the MLT and can provide large-scale wave structures that could also be traced into the thermosphere (Funke et al, 2010). Measurements by the SCIAMACHY instrument have been analysed in terms of MLT planetary wave structures in noctilucent clouds (von Savigny et al., 2007). On the Aura satellite, microwave limb measurements

by the MLS instrument have provided mesospheric planetary wave data based on composition, temperature and Doppler wind analysis (Limpasuvan et al., 2005; Wu et al., 2008). The above limb sounders can provide vertical retrieval resolutions down to a few kilometers. However, all gravity wave analysis from these limb measurements suffers from sparse horizontal sampling and the long line-of-sight integration, largely restricting the analysis to horizontal wavelengths exceeding several hundred kilometres.

Complementary to the above limb datasets, nadir-viewing satellite instruments have been analysed in terms of waves in the MLT, primarily focusing on horizontal wave structures. These studies employ observations of either noctilucent clouds or airglow layers. Basic wave parameters have been inferred from NLC observations by the UVIST instrument onboard the MSX satellite (Carbary et al., 2000). On the AIM satellite, the CIPS instrument has provided comprehensive gravity wave



information from near-nadir imaging of NLC, resulting in wave climatologies covering horizontal wavelengths both above 100 km (Rusch et al., 2008; Chandran et al., 2009) and below 100 km (Rong et al., 2018). MLT gravity wave analysis based on nadir nightglow observations has been reported from the VIIRS instrument onboard the NOAA/NASA Suomi satellite (Yue et al., 2014; Miller et al., 2015), and the IMAP/VISI instrument on-board the International Space Station (Perwitasari et al.,

2016). All gravity wave analysis from these nadir imagers is largely restricted to information about horizontal wavelengths in the observed layer. For NLC, however, information about vertical structures has recently been obtained by applying tomographic methods to the different viewing angles available from the AIM/CIPS observations (Hart et al., 2018).

For a more complete gravity wave analysis, three-dimensional retrievals are desirable that provide both horizontal wavelengths extending below 100 km and vertical wavelengths extending below 10 km (Preusse et al., 2008). Being limited

by either the limb or nadir geometry, we are so far lacking such three-dimensional gravity wave data in the MLT. In the stratosphere, techniques have been developed to overcome these limitations. The AIRS instrument on-board the AQUA satellite measures upwelling radiation in a large number of spectral channels, and gravity waves have retrieved from the upper troposphere to the mid-stratosphere (Hoffmann and Alexander, 2009; Gong et al, 2012). Additional techniques have been developed to maximize horizontal wave information from AIRS, utilizing either across-track nadir scans (Wright et al., 2017)

or data from multiple satellite tracks (Ern et al., 2017). Enhanced horizontal information can also be obtained by combing data from several sounding instruments, like multiple GPS radio occultations (Wang and Alexander, 2010; Schmidt et al., 2016), or combined data from the HIRDLS instrument on-board the Aura satellite and radio occultations (Alexander, 2015).

The ultimate way to obtain three-dimensional information about gravity waves is to apply tomographic techniques. Tomographic retrievals have been applied to the limb-scanning instruments MIPAS on-board ENVISAT (Carlotti et al., 2001;

Steck et al., 2005) and MLS on-board Aura (Livesey et al., 2006). In the mesosphere, tomographic retrieval has been applied to the limb-scanning Odin satellite to study NLC by the OSIRIS optical spectrograph (Hultgren et al., 2013; Hultgren and Gumbel, 2014), and water vapour and temperature by the SMR microwave instrument (Christensen et al., 2016). For instruments specifically designed for tomography, limb imaging is preferable over the above limb scanning techniques. In this way, the number of lines of sight through a given atmospheric volume can be maximized. This has been utilized by the infrared

limb imager of the OSIRIS instrument on-board Odin (Degenstein et al., 2003, 2004), and by the airborne GLORIA instrument (Ungermann et al., 2011; Kaufmann et al., 2015). Limb imaging also opens for a transition from two-dimensional to fully three-dimensional tomography. Ungermann et al. (2010) investigated requirements for gravity wave retrievals in the troposphere and stratosphere, emphasising the need for a fully three-dimensional tomographic analysis. Krisch et al. (2017) discussed tomographic retrieval with special emphasis on the limited range of observation angles that are typically available

from limb measurements.

In this paper, we describe a new satellite mission aiming at three-dimensional tomographic studies of gravity waves and other structures in the upper mesosphere and lower thermosphere. The MATS satellite will perform limb-imaging of the $O_2$ Atmospheric Band airglow in the near-infrared and of NLC in the ultraviolet. In combination with the tomography, spectroscopic techniques will be applied to infer atmospheric temperature and composition from the $O_2$ emissions, and



microphysical cloud properties from the NLC measurements. A complementary camera will provide nadir imaging of structures in the $O_2$ Atmospheric Band nightglow on smaller spatial scales. A similar mission with focus on the $O_2$ Atmospheric Band nightglow has recently been described Song et al. (2017). As compared to the pure limb imaging by MATS, Song et al. envisage tomographic retrievals utilizing both limb and sub-limb viewing.

The next section describes the basic ideas of the MATS mission, with focus on scientific objectives and resulting instrument requirements. Section 3 provides details about the instrument design. Section 4 introduces the retrieval ideas behind MATS and the basic data processing. Section 5 describes operational planning. Section 6 concludes with a summary and some perspectives towards scientific collaboration. Note that the idea of this paper is to provide a general overview over the mission. More comprehensive details about instruments, retrieval methods and scientific analysis will be published in separate papers.

**2 The MATS satellite mission**

**2.1 Scientific Objectives**

The primary goal of MATS is to determine the global distribution of gravity waves and other structures in the MLT over a wide range of spatial scales. Primary measurement targets are airglow in the $O_2$ Atmospheric band and sunlight scattered from NLC. These emissions will be measured in an altitude range 75-110 km, and are to be analysed in terms of wave structures
with horizontal wavelengths from tens of kilometres to global scales, and vertical wavelengths from 1 to 20 km. Over a period of two years, MATS will thus build up a geographical and seasonal climatology of wave activity in the MLT. This database will then be the starting point for scientific analysis in various directions. Relating back to the overview in Section 1, major scientific questions are:

- What gravity wave spectra are present in the MLT, and how are these related to tropospheric sources and circulation
20       conditions in the lower and middle atmosphere? These questions are tightly connected to wave-wave interactions such as filtering by planetary wave activity and in-situ generation of secondary waves.

- To what extent does MLT wave activity affect processes in the thermosphere and ionosphere? As part of this objective, methods need to be developed that utilize the mapping of mesospheric wave activity as an input to studies of thermospheric variability.

- How can explicit and parameterized implementations of gravity waves be improved in in atmospheric models? This relates back to the quest to reduce large uncertainties in current descriptions of wave sources, wave propagation, and wave interactions.

While the MATS gravity wave climatology will be the starting point for addressing these questions, complementary input from other sources will be important. This includes in particular meteorological reanalysis data, ionospheric monitoring systems,
and other dedicated missions that provide data beyond the altitude range of the MATS measurements.


As described above, NLC are one of the measurement targets of MATS. Since the pioneering days of NLC research, these clouds have transformed from a basic research object to a valuable research tool when it comes to remote sensing of the state of the MLT. Nonetheless, beyond using NLC as a convenient tracer for gravity wave studies, MATS will also address basic science questions concerning NLC in their own right:

•   How are NLC affected by gravity waves and other transient processes in the MLT? This concerns both the microphysics of ice particles and the resulting evolution of observable cloud structures.

## 2.2 Measurement concepts

The above scientific objectives define the requirements on the measurements that MATS will perform. Tomography is the
basis for obtaining three-dimensional information on spatial scales that are relevant for gravity wave studies. The tomographic retrieval needs input in terms of multiple line-of-sight observations through a given atmospheric volume. This is achieved by a limb imager that observes the atmosphere along the Earth's tangent direction, with a field of view covering tangent altitudes between 75 and 110 km, and 300 km across track. Figure 1 illustrates the observation geometry. In order to tomographically retrieve gravity wave information in the MLT, we need to utilize atmospheric emissions that are both sufficiently bright and
susceptible to gravity wave activity. In the case of MATS, we utilize airglow in the $O_2$ Atmospheric Band and scattering of sunlight by NLC. Both $O_2$ Atmospheric Band airglow and NLC feature horizontal and vertical structures that are a direct response to gravity wave activity. As an additional benefit, both phenomena allow for a deeper analysis by applying spectroscopic techniques.

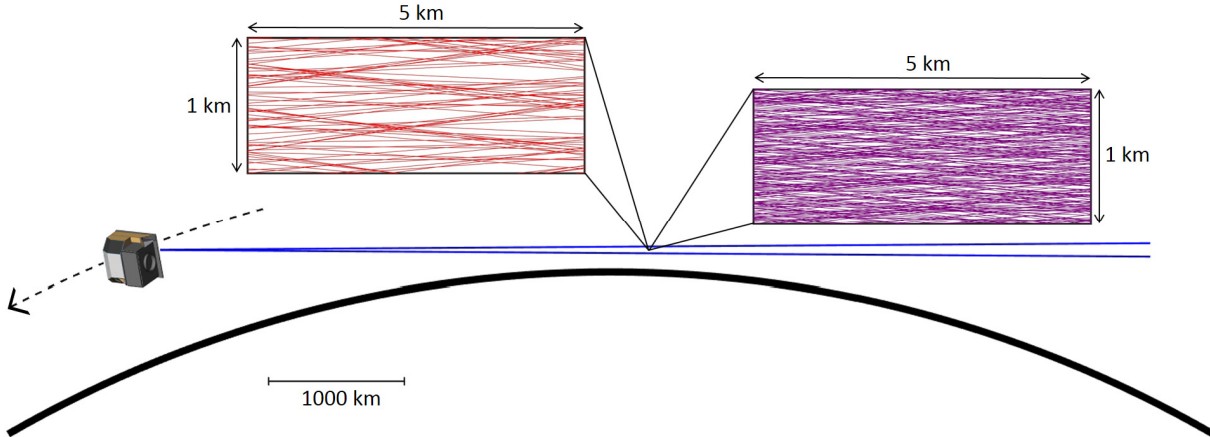

**Figure 1. Illustration of the MATS limb observation geometry. From an orbit altitude of 585 km, the vertical field of view of the limb instrument covers nominal tangent altitudes from 75 to 110 km. The smaller inlays show examples of lines-of-sight filling a section of 5×1 km in the orbit plane. The density of the lines-of-sight is representative for the binned image pixels in an $O_2$ Atmospheric Band channel (left) and an NLC channel (right).**



On MATS, the $O_2$ Atmospheric Band is measured using four spectral channels in the near infrared between 750 and 775 nm (see Table 4 for details). Measurements will be performed both during daytime (dayglow) and nighttime (nightglow). As a primary step, the tomographic analysis will convert measured limb radiances to volume emission rates. Combining the four channels, subsequent spectroscopic analysis will utilize the rotational structure of the $O_2$ Atmospheric Band emission to infer temperature (Babcock and Herzberg, 1948; Sheese et al, 2010). At the same time, the total volume emission rate in the $O_2$ Atmospheric Band can be analysed in terms of odd oxygen densities: The Atmospheric Band nightglow provides a direct measure of atomic oxygen density (McDade et al., 1986; Murtagh et al., 1990). The Atmospheric Band dayglow, on the other hand, provides information about ozone density (Evans et al., 1988; Mlynczak et al, 2001), which in turn is related to atomic oxygen through photochemical equilibrium. Gravity waves in the MLT can be inferred from these measurements by observing patters in either airglow volume emission, odd oxygen, or temperature. Among these, analysing gravity waves in the temperature field is most beneficial as temperature is directly connected to the basic state of the atmosphere and as gravity wave momentum flux becomes accessible (Ern et al, 2004; Fritts et al., 2014). To this end, temperature amplitudes as well as horizontal and vertical wavelengths need to be inferred from the measurements.

NLC are measured by MATS using two spectral channels in the ultraviolet at 270 and 305 nm. While imaging at one wavelength is sufficient for analysing global NLC variations and local NLC structures, the use of two wavelengths gives the additional benefit of accessing particle sizes and ice content. To this end, the observed spectral dependence of NLC signal is fitted in terms of an Ångström exponent and compared to numerical scattering simulations (von Savigny et al, 2005; Karlsson and Gumbel, 2005). For typical NLC particle sizes, observations in the ultraviolet are preferable as they push the scattering deeper into the Mie regime, thus maximizing the amount of information that can be inferred from spectral measurements. In addition, wavelengths below 310 nm are efficiently absorbed by the stratospheric ozone layer, and are therefore chosen to avoid complications due to upwelling radiation. For MATS, the concrete wavelengths 270 nm and 305 nm are chosen both to ensure a sensitive retrieval in the NLC particle size range of interest (Section 4.6), and to minimize potential perturbations due to atmospheric emission features (airglow, aurora). The tomographic NLC data will be the basis for gravity wave analysis in terms of horizontal wavelengths. The vertical structure of the NLC is strongly determined by the microphysics that governs cloud growth and sedimentation (Rapp and Thomas, 2006). The tomography will provide detailed insights in this vertical NLC evolution, including its possible modification by wave activity (Hultgren and Gumbel, 2014; Megner et al., 2016; Gao et al., 2018). However, since the vertical structure of the narrow NLC layers is dominated by microphysics rather than dynamical processes, a retrieval of vertical wavelengths of gravity waves will not be feasible from the NLC data.

The limb instrument will be described in Section 3, and details of the tomographic and spectral retrievals will be given in Section 4. Table 1 summarizes the above retrieval products from the MATS limb measurements. The table also states the required precision and spatial resolution of these retrieval products. These requirement are defined by the need to infer relevant gravity wave data from these retrieval products, in accordance with the overall objectives listed in the Section 2.1. Note that typical values are given for precision and resolution. These parameteres depend on the altitude-dependent signal strengths.





They are also adjustable as e.g. enhanced image binning can improve precision at the cost of resolution. These trade-offs will be further illustrated in the following sections.

| | O$_2$ Atmospheric Band dayglow/nightglow | | | NLC | |
|---|---|---|---|---|---|
| | **Emission rates** | **Temperature** | **O / O$_3$ abundance** | **Brightness** (scattering coefficient) | **Particle sizes** (Ångström exponent) |
| Temporal coverage | all seasons | all seasons | all seasons | summer | summer |
| Geographical coverage | global | global | global | poleward of 45° | poleward of 45° |
| Altitudes | 75-110 km | 75-110 km | 75-110 km | 80-86 km | 80-86 km |
| Precision | 1-5 % | 2-5 K (day) 5-20 K (night) | 1-5 % | 2-5 % | 0.25 |
| Retrieval resolution (along track × across track × vertical) | 60×20×1 km | 60×20×1 km | 60×20×1 km | 60×10×0.5 km | 60×10×0.5 km |

**Table 1: Products of the tomographic/spectroscopic retrievals from the MATS limb measurements. These serve as input to subsequent wave analysis.**

In addition to these MATS limb measurements, a nadir imager will take pictures of the O$_2$ Atmospheric Band emission from below the satellite. This provides complementary information on smaller spatial scales down to 10-20 km horizontal resolution, albeit restricted to a detection of structures rather than a detailed spectroscopic analysis. During sunlit (or moonlit) conditions, nadir measurements of airglow layers get drowned in background light from the lower atmosphere. Gravity wave data from the MATS nadir camera will therefore largely be restricted to moonless nighttime conditions. From the ground, nightglow imaging is a standard technique for local measurements of gravity waves in the MLT [e.g., Taylor et al., 1997; Espy et al., 2004]. An obvious advantage of satellite measurements is global coverage, however, this comes with the disadvantage of lacking temporal coverage at a given location. Also, nadir imaging from a moving satellite is subject to image smearing (motion blur), thus implying a restriction to strong features and short integration times.

**2.3 Mission development**

Original ideas for MATS date back longer than the current project development. A first mission concept was developed by Jacek Stegman and Donal Murtagh at Stockholm University in the 1990s, then under the name "Mesospheric Airglow Transient Signatures (MATS)". An important heritage for MATS is also the Odin satellite mission, both concerning satellite, instrument, and operational concepts (Murtagh et al., 2002; Lelwellyn et al., 2004). For the Optical Spectrograph and InfraRed Imager System (OSIRIS) on-board Odin, tomographic ideas were developed by Degenstein et al. (2003; 2004). In 2010, special "tomographic" scan modes were developed for Odin, covering a limited tangent altitude range of about 75-90 km with



relatively high horizontal repetition rate. These measurement provided input to tomographic and spectroscopic retrievals (Hultgren and Gumbel, 2014) that served as important tests for the MATS mission development.

The current MATS satellite mission was developed in response to a call by the Swedish National Space Agency concerning "Innovative low-cost research satellite missions". MATS was selected after going through an initial Mission Definition Phase
in 2014.

An important basis for MATS is the InnoSat satellite platform developed by OHB Sweden and ÅAC Microtec (Larsson et al., 2016). InnoSat has been designed as a "universal" microsatellite platform that can host a variety of different payloads for aeronomy or astronomy research in low-earth orbit. MATS is the first scientific mission to use InnoSat. As a consequence, much of the development of platform and payload have been carried out in parallel. MATS has been designed to use the
"baseline configuration" of InnoSat. Table 2 lists important parameters that define this configuration. All parameters in Table 2 constitute boundary conditions for the design and performance of MATS, as will be illuminated further in Sections 3 and 4.

| Mass | 50 kg  (incl. 20 kg payload) |
|---|---|
| Size | 85×70×55 cm |
| Power | 45 W on orbit average |
| Data volume | 180 MBytes per day |
| Pointing accuracy (in terms of limb altitude) | 5 km pointing error (target) 0.5 km knowledge error (reconstruction) |
| Orbit | sun-synchronous, near-terminator, polar orbit |
| Nominal lifetime | 2 years |

**Table 2: Selected parameters of the InnoSat satellite platform in its baseline configuration.**

Behind the development of the MATS instruments and scientific mission is an Instrument Consortium comprising Stockholm University, Calmers University of Technology in Göteborg and the Royal Institute of Technology in Stockholm, in collaboration with Omnisys Instruments (Göteborg), Molflow (Göteborg), and Kyung Hee University (Republic of Korea). This is complemented by a Platform Consortium comprising OHB Sweden in Stockholm and ÅAC Microtec in Uppsala, the
companies behind InnoSat. The MATS satellite is currently in preparation for a launch in late 2019. The launch will take place in a piggyback configuration on a Soyuz-2-1b rocket from Vostochny in Eastern Russia. The nominal altitude of the orbit is 585 km. The nominal local time of the equator passage is 6:30 and 18:30, thus providing a "near-terminator" orbit.



## 3 Instrument Design

### 3.1 Overview

The MATS payload comprises four optical instruments: The 6-channel limb imager and the nadir camera measure mesospheric emissions, as introduced in the previous section. A pair of nadir-viewing photometers measures upwelling radiation from the Earth surface and lower atmosphere, in support of the limb instrument analysis. A star-tracking camera directed in the opposite direction of the limb imager ensures accurate pointing of the satellite. Figure 2 shows the overall configuration.

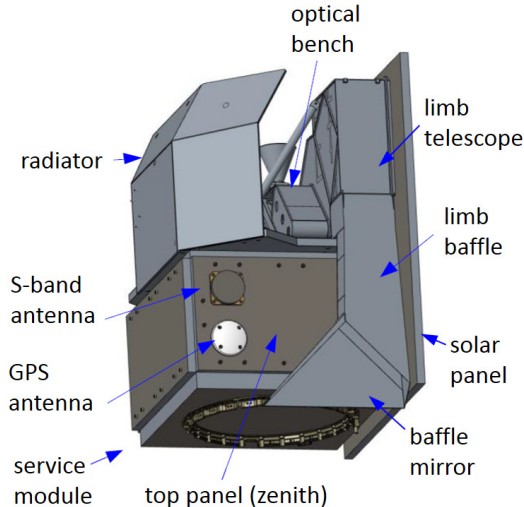

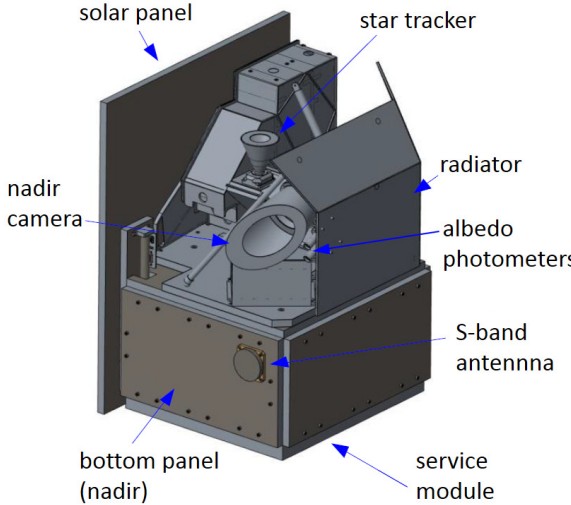

**Figure 2. Layout of the MATS satellite.**





The sun-synchronous polar orbit with nominal equator passage near 06:30 and 18:30 local time is beneficial for an efficient satellite design. It ensures that satellite receives sunlight largely during the entire mission, and that a solar panel mounted at one side of the platform is sufficient to make use of this sunlight. In addition, instruments and electronics will be shaded behind

the solar panels during the mission. As for the field of view, the satellite will nominally be oriented so that the limb instrument looks backwards along the orbit. This provides the necessary overlap between subsequent images as input to the tomography retrieval.

## 3.2 Limb instrument

Since the goal is to investigate both NLCs and $O_2$ Atmospheric Band emission, two separate wavelength regions will be measured by MATS. As described in Section 2.2, two UV channels will be used for the NLC study. To measure the Atmospheric Band emission, two main channels are used: a wideband channel covering the entire 0-0 vibrational band, and a narrowband channel covering only the centre. In order to quantify the effect of background radiation and straylight on the Atmospheric Band measurements, a set of ancillary measurements are done using two background IR channels of the limb

imager, as well as the pair of nadir-looking photometers that provide information about upwelling radiation both within and outside the Atmospheric Band. In order to achieve the MATS measurement objectives (Table 1), four basic tasks have been central to the limb instrument design: imaging quality, sensitivity (signal-to-noise ratio), spectral separation, and straylight suppression. Boundary conditions for the design are defined by the InnoSat satellite platform in terms of mass, dimensions, power etc. (Table 2). Figure 3 shows the overall layout of the limb imager.

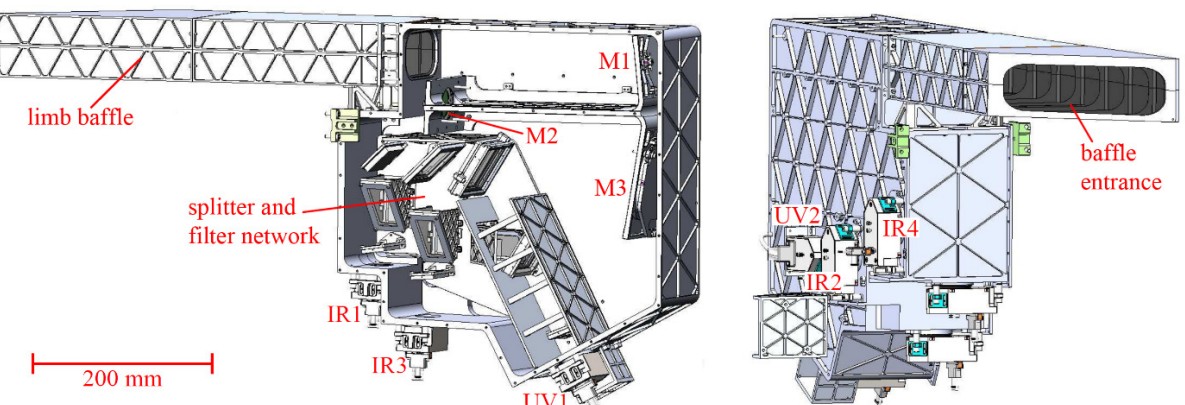

**Figure 3. Overview of the MATS limb imager with one of the side covers removed (Hammer et al., 2018). Marked in the image are the telescope mirrors M1-M3, as well as the CCDs IR1-IR4 and UV1-UV2 for the six spectral channels.**





### 3.2.1 Telescope

The limb instrument is based on a single off-axis three-mirror reflective telescope (f/D = 7.3) with a field of view (FOV) of 5.67° × 0.91°. The mirrors are manufactured by Millpond ApS with fully free form surfaces. They are made of aluminium with the active surfaces defined using diamond turning. The free form design was optimized to achieve diffraction-limited

imaging. Inter-mirror distances and angles were chosen to satisfy the linear-astigmatism-free condition. Linear astigmatism is the dominant aberration of off-axis reflecting telescopes and must be eliminated to obtain a wide field of view (Chang, 2015). A summary of properties of the limb telescope is found in Table 3.

| Type | Linear-astigmatism-free off-axis three-mirror reflective |
|---|---|
| Mirrors | diamond turned aluminium with protective UV coating, 3 nm rms |
| Field of view | 5.67° × 0.91° (250 km × 40 km) |
| Entrance pupil | 9.6 cm$^2$ |
| Focal length | 260 mm |
| f / D | 7.3 |
| Aperture stop | located on the secondary mirror |

**Table 3. Overview of optical properties of the MATS limb telescope.**

### 3.2.2 Splitter and filter network

Following the telescope is a network of dichroic beam splitters and thin-film interference filters that are used to achieve the desired spectral selection. As the first element, a beamsplitter BS-UV-IR reflects wavelengths below 345 nm towards the UV

part of the instruments, while longer wavelengths are transmitted towards the infrared part. Figure 4 shows the detailed distribution of spectral channels and optical elements. Each of the instrument's six channels uses a broadband filter to remove out-of-band signals, followed by a narrowband filter that ultimately defines the wavelengths transmitted to the image sensors. In addition, two folding mirrors are used to keep the optical components within the InnoSat platform envelop. Optical tests performed at breadboard and prototype level shows that the resolution requirement for the IR channels are fulfilled, while

more careful mirror alignment is needed for the flight model to meet the imaging requirements of the UV channels.

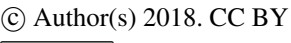



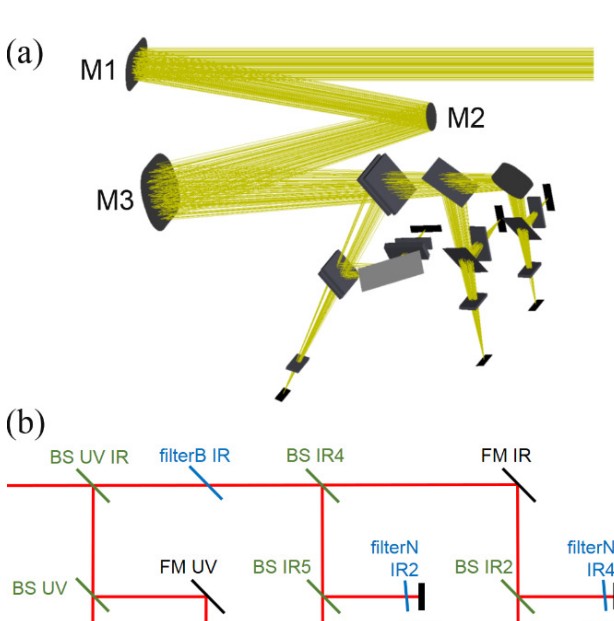

**Figure 4. Overview of the optical paths in the limb imager. (a) Geometrical distribution of the optical elements, including the three telescope mirrors M1-M3. (b) Schematic of the channel layout with beamsplitters (BS), broad filters (filterB), narrow filters (filterN), folding mirrors (FM), and CCDs. See also Figure 3.**

| Designation | Central wavelength | Band-width | Tangent altitudes | Resolution (Imaging) | Resolution (Binning) | Signal/Noise Ratio |
|---|---|---|---|---|---|---|
| UV1-short | 270 nm | 3 nm | 70-90 km | 0.2 km | 0.2×5 km | 100 |
| UV2-long | 304.5 nm | 3 nm | 70-90 km | 0.2 km | 0.2×5 km | 100 |
| IR1-ABand-centre | 762 nm | 3.5 nm | 75-110 km | 0.4 km | 0.4×10 km | 500 |
| IR2-ABand-total | 763 nm | 8 nm | 75-110 km | 0.4 km | 0.4×10 km | 500 |
| IR3-BG-short | 754 nm | 3 nm | 75-110 km | 0.8 km | 0.8×50 km | 500 |
| IR4-BG-long | 772 nm | 3 nm | 75-110 km | 0.8 km | 0.8×50 km | 500 |
| Nadir | 762 nm | 8 nm | nadir | 10 km | 10×15 km | 100 |

**Table 4. Overview of optical properties of the MATS limb and nadir channels. The image resolution of the channels is specified in two ways: Imaging refers to the imaging quality of the optics in terms of the full width at half maximum of the point spread function.**
10 **Binning refers to the size of the recorded image pixels after nominal binning on the CCD. Pixel sizes are given as vertical×horizontal for the limb channel, and as across track × along track (including motion blur) for the nadir channel.**

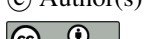



### 3.2.3 Baffle design

One of the major design drivers for the limb instrument has been to minimize the impact of straylight from outside the field of view. This task is critical considering that the bright lower atmosphere is only 1-2° below the nominal mesospheric field of view. Central to the straylight handling is a long (>650 mm) baffle in front of the primary telescope mirror. To minimise the

reflections inside the baffle, it is coated internally with Vantablack S-VIS, which has a reflectivity of less than 0.6% in the wavelength regions relevant for MATS. Furthermore, the limb housing and all mounting structures are coated using a black nickel (Hammar et al., 2018). During most of the mission, the baffle entrance will be in the shadow of the solar panel. However, during some high-latitude summer conditions, the sun can illuminate satellite structures near the baffle. In order to minimize the risk of straylight entering the baffle, a plane mirror is placed in front of the baffle entrance (Figure 2). As opposed to (black)

surfaces that can scatter incident light in uncontrolled ways, this "baffle mirror" has been designed to reflect sunlight away from the instrument.

Since the MATS limb telescope lacks a field stop, Lyot stops are used in front of

each image sensor. In addition, all sensors are deeply embedded in the structure. These measures ensure that the critical paths from the primary and secondary mirrors are removed. Furthermore, the inter-mirror distances were chosen to be as large

as possible while still fitting into the available payload volume. By doing so, the subtended angles between the mirrors were minimized, which, in turn, minimizes the throughput of scattered light emanating from outside the nominal field of view.

To verify the performance of the stray-light suppression, a combination of experimental testing and modelling of the instrument in Zemax Opticstudio have been carried out (Hammar et al., 2018). From this, attenuations better than $10^{-5}$ are generally obtained for angles exceeding 1.5°, thus fulfilling the requirements for the mission.

### 3.2.4 Readout electronics

To record the incoming light, all channels use passively cooled back-lit CCD sensors (Teledyne E2V-CCD42-10). The data from each CCD are read out by a CCD Readout Box (CRB) with an instrument onboard computer (OBC) to handle the data. Two power and regulation units, located in the instrument electronics box together with the OBC, provide adjustable voltages for CCD operation, and multiplex the control of the CRB settings and the data readout, for up to four imaging channels. The

OBC then compresses the image (if applicable) before handing over to the Innosat platform which manages the satellite downlink. The nominal image format will be compressed 12-bit JPEG images, while full resolution uncompressed image readout is also available for in-flight calibration purposes.





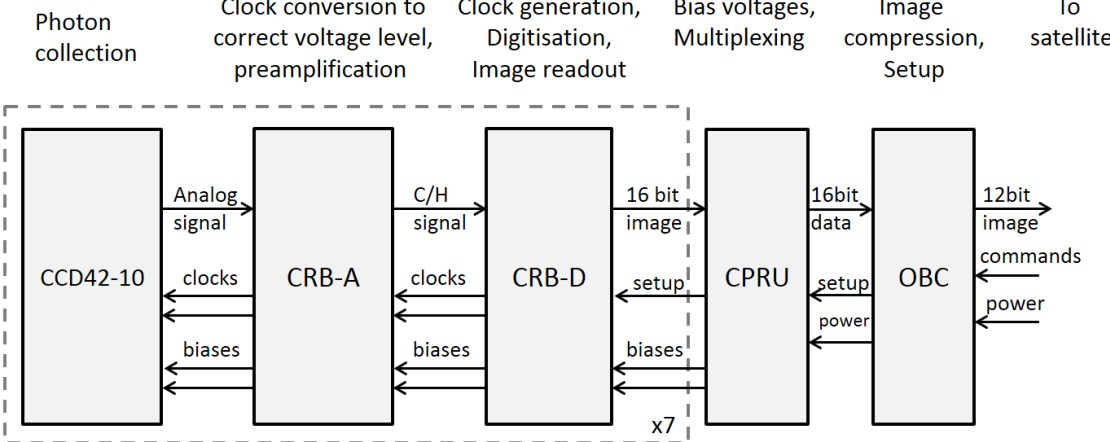

**Figure 5. Overview of the MATS image acquisition. CCDs are controlled and read out by CCD Readout Boxes (CRB-A and CRB-D), connected to the CCD power regulation unit (CPRU) and onboard computer (OBC).**

The CCD provides 512x2048 image pixels. The field of view of each limb channel occupies a wide (along the limb) and short (in the vertical direction) area of interest. CCD readout implies vertical shift of the image rows. To minimize the number of moving parts in the satellite no shutter is used in the instrument. As a consequence, the image rows continue being exposed during the readout shifting, resulting in image smearing. The effect of this is minimized by fast readout (using binning and

10 skipping rows outside the region of interest) as well as correcting for smearing in post-processing (Section 4.2).

To minimize noise and interference in CCD readout, the read-out electronics is composed of two parts, analog and digital. The analog box (CRB-A) is located in immediate vicinity of the CCD. The function of CRB-A is to generate the necessary clock signals for the CCD and to provide signal conditioning for the CCD output signal. The clock signals are generated in the digital box (CRB-D, located together with the rest of the instrument electronics) with standard logic voltage level, and are

15 converted to the voltages needed by the CCD inputs by dedicated gate drivers. The signal from the CCD is pre-amplified and handled by a clamp and hold circuit. The amplified analog signal is passed over a differential connection to the CRB-D, where it is digitized by a 16-bit Analogue-to-Digital Converter (ADC) and stored in memory, available for transfer to the OBC. CRB-D uses a Field-Programmable Gate Array for generating the multiple clocks for the CCD, and sending the image to the OBC.

Since the different MATS channels and science modes have different requirements on the final image, the CRB firmware

is flexible, allowing changing multiple settings of the readout, such as integration time, region of interest on the CCD, horizontal and vertical binning, or CCD output amplifier selection. An overview of the planned settings for the nominal science modes will be provided in Section 0



Exposure of the CCD to radiation in orbit will affect the dark current performance, which can be counteracted by adjusting the bias and clock voltages for the CCD in the power regulation unit. Hot pixels may develop on the CCD due to radiation. These can be excluded from binning, by flagging the columns that contain them as bad.

Giono et al. (2017) carried out performance measurements on a prototype version of the readout electronics, showing readout noise of about 50 e$^-$ per CCD pixel using the high signal mode amplifier on the CCD. This can be further reduced to under 20 e$^-$ per pixel by adjusting the pre-amplification gain and by using the low signal mode amplifier.

### 3.3 Albedo photometers

Each of the two albedo photometers consists of a two-lens telescope based on standard N-BK7 lenses, providing a field of view of 6°. In front of the telescope a pair of interference filters is placed on each photometer to discriminate against unwanted wavelengths. Baffles minimize straylight from outside the field of view. The detector is a Hamamatsu S1223-01 Si PIN photodiode. The system measures upcoming radiation at 759-767 nm and 752.5-755.5 nm, corresponding to the limb imager channels IR2-ABand-total and IR3-BG-short, respectively (Table 4). The signal to noise ratio better than 100.

### 3.4 Nadir camera

The nadir camera is a Cooke triplet with an entrance pupil of 15 mm, and an effective focal length of 50.6 mm. Its field of view is 24.4° × 6.1°. From orbit, this covers an area of at least 200 km across track and 50 km along track at 100 km altitude. The design can resolve 10 × 10 km features at this altitude. Additional degradation occurs in the along-track direction due to smearing caused by the satellite movement during the exposure and read-out phase. Figure shows a sample image taken by a prototype nadir camera from the ground. In orbit, the nadir measurements aim at nighttime conditions with the Sun located at least 10° below the horizon for a ground-based observer. However, under these conditions the satellite will be fully illuminated by the Sun, and straylight handling is thus essential for the nadir camera. Similar to the limb instrument, in addition to being mounted in the shadow of the solar panel, efficient straylight suppression for the nadir camera is achieved by a baffle with black coating on the inside.



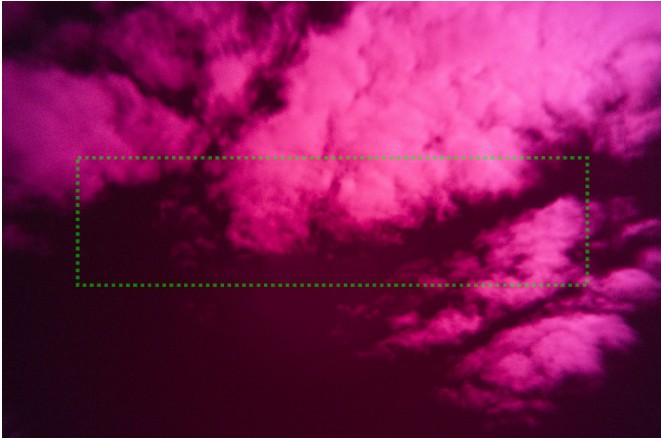

**Figure 6. Ground-based photograph of a cloudy sky taken with the nadir objective mounted on a Canon camera house with a full frame sensor (35 mm). The dashed green rectangle denotes the cropped field of view as it will be seen with the MATS CCD sensor.**

The readout of the nadir camera uses the same readout electronics as the limb instrument (Section 3.2.4), albeit operated in a different readout scheme. In order to limit motion blur, nightglow images are taken with an exposure time no longer than 1 s. Exposures are taken and the CCD is read out at a rate sufficient to obtain overlapping images along the ground-track of the satellite. The result is a continuous nightglow image swath of width 200 km along the night-part of the orbit.

**4 Data processing**

**4.1 Overview**

The data produced by the instruments on board MATS requires several processing steps. The three major steps are:

Level 0:   Geolocating the images and adding meta-data relevant for further processing.

Level 1:   Calibrating the images (and photometer measurements) such that the pixel values reflect the actual measured
radiance.

Level 2:   Linking those values to the physical properties of the atmosphere via the 3D tomographic reconstruction and
spectroscopy.

Since the first step is mainly an administrative step for further processing, only the Level 1 and 2 processing will be discussed in this paper. Focus will be on the limb instrument as the main instrument on MATS. Figure 7 shows the overall processing
chain for the limb data.



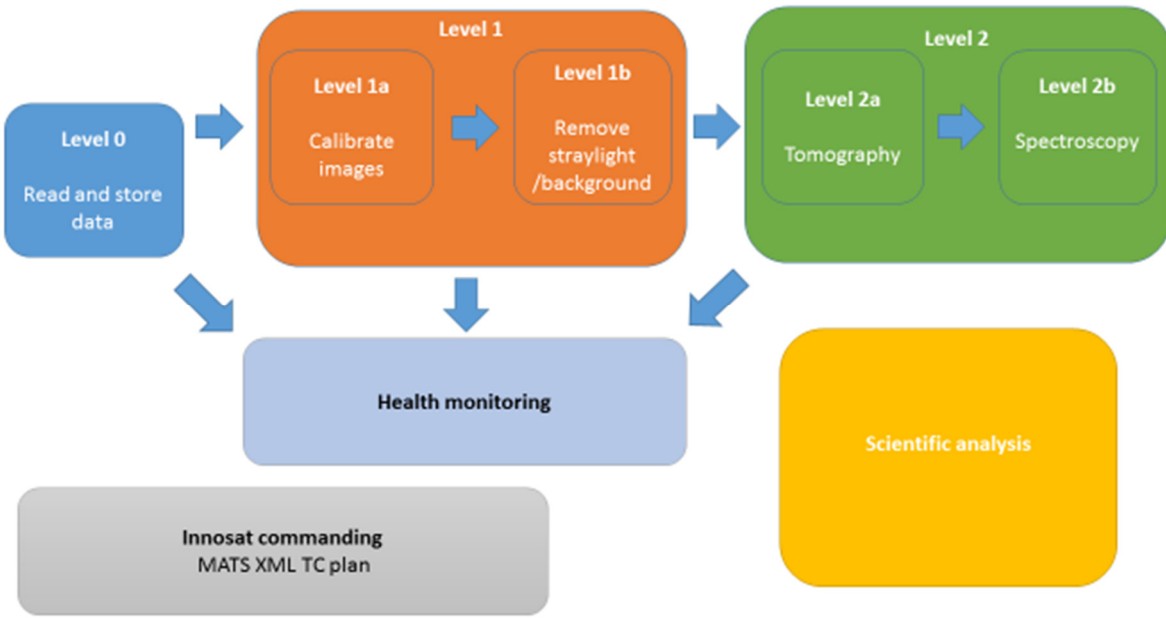

**Figure 7. Overview of the data processing steps for the MATS limb imager.**

## 4.2 Calibration of images

After geolocation and time-tagging, each image from the limb channels must be calibrated such that the image displays the radiance falling on each binned CCD pixel. The main effects that need to be accounted for during the calibration of the CCD images are: 1) biases in the readout electronics, 2) readout smearing, 3) dark current, and 4) flat field correction.

Each of these effects behaves differently, and must be measured and modelled separately. The parameters used in this modelling are a combination of pre-flight calibration measurements, and in-flight corrections to these parameters via special calibration modes (see Section 5.2).

### 4.2.1 Readout bias

The bias produced by the readout electronics is determined for each image. This is done by means of blank pixel values, i.e. the signal from unexposed pixels in the readout register of the CCD. These pixels are shifted each time a row is read out, thus accumulating very little charge between consecutive rows. Two values, representing the average of the leading and trailing blank pixels, respectively, are recorded with each image as auxiliary information.





### 4.2.2 Readout smearing

In order to avoid risks associated with moving parts on the satellite, the CCDs have not been equipped with shutters. As a consequence, the CCD pixels are continuously illuminated, even during the image shifting associated with the readout, and are thus contaminated with signal from the "wrong" part of the image. To compensate for this effect, the image must be de-
smeared by recursively correcting the pixel values in the CCD rows. While this process is completely deterministic for ideal noise-free measurements, it introduces minor image errors in the real data.

### 4.2.3 Dark current

The thermal energy of the CCD gives rise to electrons that are not due to incoming photons, but are nonetheless captured by the CCD's potential wells and counted as signal. The strong temperature dependence of this dark current is characterized pre-
flight for the individual CCDs. In-flight measurements will monitor the detailed dark current properties of individual pixels throughout the mission.

### 4.2.4 Flat field correction

Finally, the image is corrected for variations of the sensitivity across the image. Based on pre-flight calibration, this flat field correction accounts for variations of both the optical throughput through the limb instrument and the CCD quantum efficiency.
The results is a map of the true pixel illumination.

### 4.3 Straylight and background removal

Before the limb radiances can be analysed in terms of airglow or NLC, two unwanted contributions to the signal must be removed: background originating from within the nominal field of view, and straylight originating from outside the nominal field of view. Background from inside the field of view comprises both unwanted emissions (airglow, aurora) and scattered
light (in particular molecular Rayleigh scattering). Straylight from outside the field of view can reach the CCDs by scattering in the baffle, scattering from imperfect or dusty surfaces of the optical elements (in particular the primary telescope mirror), and/or scattering from structures inside the instrument housing. Although the design of the limb instrument is optimized for out of field rejection, some straylight signal is to be expected. To remove this signal, both amplitude and non-uniformity across each CCD need to be estimated. As described in the following subsections, the amplitude will be estimated by combining
information from several channels and tangent altitudes. The non-uniformity will be parameterized based on the straylight modelling and testing carried out prior to launch (Hammar et al., 2018).

Based on the measurement data in orbit, the contributions of background and straylight to the total signal can be difficult to distinguish from each other. In addition, both wanted and unwanted signals can be expected to vary in similar ways along the orbit. Most notable, upwelling radiation, and thus local conditions at lower altitudes, will affect $O_2$ Atmospheric Band
dayglow, NLC scattering, molecular Rayleigh background and straylight. In the MATS data processing, the procedures for



removing background and straylight are therefore linked and will partly rely on the same auxiliary information, e.g. from the IR background channels or the albedo photometers.

### 4.3.1 Airglow channels

The two IR background channels (Table 4) provide the starting point for removing background and straylight in the $O_2$ Atmospheric Band channels. The amplitude of the straylight is estimated from the signals seen in the IR background channels at the highest altitudes, where we expect negligible contribution from other sources of light. The background from the nominal feild of view, on the other hand, is estimated using the full background channel images. It compensates not only for the Rayleigh background, but also for the possible presence of NLC, and for airglow and possibly auroral emission features. However, the total contribution by these signals cannot be estimated by simple linear interpolation between the two IR background channels. Rather, these signals show distinct spectral dependence (Sheese et al., 2010). In particular, since $O_2$ resonantly absorbs upwelling radiation in the Atmospheric Band itself, both scattered background and straylight are weaker than what would be expected from linear interpolation between the two background channels. To account for this, the lower atmospheric albedo needs to be quantified, regarding both absolute albedo and relative flux inside and outside the Atmospheric band, which in turn depends on lower atmospheric cloudiness and cloud top height. This will be monitored by the pair of nadir-looking albedo photometers that measure upwelling radiation inside and outside the Atmospheric Band onboard MATS. In combination with radiative transfer simulations of the relevant processes (Bourassa et al., 2008), this provides a more quantitative straylight and background correction, following the method described by Sheese et al. (2010).

### 4.3.2 NLC channels

Signals to be removed from the NLC data are molecular Rayleigh background and straylight. Because of efficient absorption of UV radiation by stratospheric ozone, the major source for out-of-field straylight is Rayleigh scattering in the upper stratosphere, and the amount depends on the atmospheric ozone abundance and solar position. Similar to the IR, the amplitude of straylight will be estimated by assuming that the signal seen at the highest altitudes in the NLC images is completely dominated by straylight. As an option, Rayleigh background and the straylight will be removed only after the tomography has been completed. This is done to combine information from both spectral channels and from as many images as possible to estimate the Rayleigh scattering and straylight, which can be assumed to vary slowly in the horizontal direction. Atmospheric density profiles from an atmospheric model (MSISE-90) will provide an initial estimate of the molecular Rayleigh background in the field of view. This estimate can then further be improved by normalizing it to the scattering observed under cloud-free conditions.





## 4.4 Tomography

### 4.4.1 Terminology

Tomography will be applied on the images to reconstruct three-dimensional fields of atmospheric emission (or scattering). This will be done using an iterative maximum a-posteriori method (MAP) (Rodgers, 2000). Here the 3D field of emission is

described by a state vector, $x$, and the measured limb radiances by the measurement vector, $y$. These vectors are related via a linear forward model

$$y = K x \qquad (1)$$

$K$ is the Jacobian matrix and describes how emissions $x$ from locations throughout the measurement volume contribute to radiances y from individual limb lines of sight. $K$ thus contains the physics of the measurement, including observation

geometry, radiative transfer, and instrument characteristics. For the MATS retrieval processing, $K$ will be geometrically calculated on a grid using a spherical (rotating) earth geometry. Absorption by ozone (in the UV channels) and self-absorption (in the Atmospheric Band channels) will be included using a pre-existing climatology to calculate the optical depth along the path.

Assuming that we have some a-priori knowledge about the atmospheric emission described by the vector $\hat{x}$ and a

covariance matrix $S_a$, the maximum a-posteriori state, $x$, can be found using Bayesian estimation by solving the equation

$$\hat{x} = (K^T S_e^{-1} K + S_a^{-1})^{-1} K^T S_e^{-1} (y - Kx) \qquad (2)$$

where $S_e$ is the covariance matrix for the measurement vector $y$.

For large scale problems, inverting equation 2 can become extremely memory-intensive to the point where a direct solver based on decomposition no longer is a possibility. Thus, the equation needs to be solved iteratively. Ungermann et al. (2010)

have shown that this can be done efficiently by rewriting the equation above as

$$(K^T S_e^{-1} K + S_a^{-1})\hat{x} = K^T S_e^{-1} (y - Kx) \qquad (3)$$

and solving it iteratively using the conjugate gradient method.

As Figure 1 illustrates, the MATS observation geometry provides a large number of lines of sight through a given atmospheric volume. However, a challenge for the MATS tomographic retrieval is that lines of sight only span over a limited

range of observation angles (6°), and that lines of sight cover very long paths (hundreds of kilometres) through an airglow layer or NLC layer. Tomographic retrievals under these conditions have been discussed by Krisch et al. (2017).

### 4.4.2 Test retrievals of NLCs

To illustrate the feasibility of the tomographic reconstruction, a prototype retrieval has been set up using a simple forward model with a pure spherical geometry (non-rotating earth) and ignoring atmospheric absorption. A three-dimensional test field

of NLC scattering coefficients is based on a combination of Odin/OSIRIS vertical profiles and AIM/CIPS images, and covers a horizontal area of 5000 km along orbit and ±175 across track (upper panel in Figure 8). Forward model simulations and retrievals are performed on a set of measurements covering roughly 3000 km along track (containing 167 limb images to be





processed), ±125 km on each side of the orbit plane, and altitudes from 60 to 100 km. The resolution of the grid is 20×6.4×0.5 km in the along-track, across-track and vertical direction, respectively. Measurements are simulated with random noise added assuming shot-noise-limited performance with signal to noise ratios defined by the instrument specification. The measurement covariance matrix is set correspondingly (diagonal elements only). The retrievals are performed with very lax constraints using

an a-priori atmosphere equal to the background atmosphere and an a-priori covariance matrix with only diagonal entries equal to $10^{-8}$ $m^{-1}$ $sr^{-1}$.

The result from this retrieval test is shown in the lower panel of Figure 8. For the area fully covered by MATS measurements, i.e. 2000-3000 km along-track / +-125 km across-track, the retrieval successfully reproduces the atmospheric field. For the area fully covered by the tomography, the mean square error amount to $1.5×10^{-10}$ $m^{-1}$ $sr^{-1}$, which corresponds to

10 a relative error of 3% for a typical cloud brightness of $5×10^{-9}$ $m^{-1}$ $sr^{-1}$. As expected, some degradation can be seen on the edges due to limited tomographic information, with no information at the outermost regions where no measurement data is available. Along-track these effects will be mitigated by performing the retrieval on subsequently overlapping volumes along the orbit.

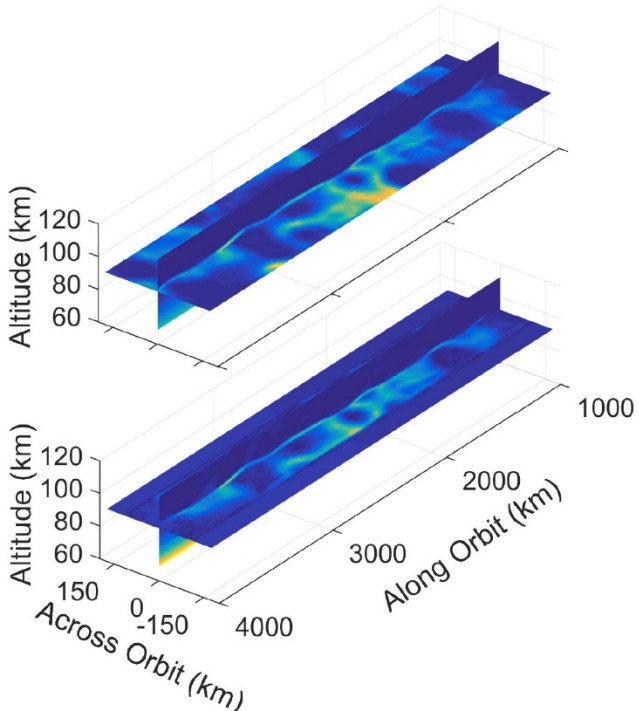

**Figure 8. Example of tomographic retrieval simulations. The upper panel shows the simulated "true" NLC volume scattering coefficient. The lower panel shows the retrieved NLC. Based on the simulated lines of sight, full retrieval is available in an area 2000-3000 km along orbit and ±125 km across orbit.**



### 4.4.3 Test retrievals of gravity waves

Since one of the major goals of MATS is to reconstruct gravity wave structures, the tomographic method has been also tested on a set of coherent gravity waves observable in the Atmospheric Band emission. The emission field is generated using a simple airglow gravity wave model (Li, 2017). Using the same forward model as for the NLC simulations, MATS limb

instrument images are simulated, and the three-dimensional emission fields are retrieved using the MAP method. 200 images have been simulated, covering 6000 km along-track and 400 km across-track with a resolution of 5×5×0.25 km.

This has been tested for wave structures aligned both along the movement of the satellite, and perpendicular to it. As part of these tests, the horizontal and vertical wavelengths are varied. The amplitude of the retrieved wave is then compared to the amplitude of the true wave field. The ratio between these indicates the contrast in the retrieved data and is referred to as the

gravity wave sensitivity for a certain wavelength (Preusse et al., 2002). In Figure 9, the sensitivity is shown for along-track waves, i.e. waves with fronts aligned perpendicular to the satellite track. Gravity waves with horizontal wavelengths down to 60 km and vertical wavelengths down to 3 to 5 km can be detected with a contrast better than 0.8. Also for across-track waves (not shown), vertical wavelengths can be inferred down to 3 km with a strong signal, but horizontal wavelengths can be retrieved down to 20 km, only limited by the resolution of the image. For both along-track and across-track waves, wave

structures with vertical wavelength down to 1 km can be detected, albeit with a reduction in amplitude by more than 50 %.

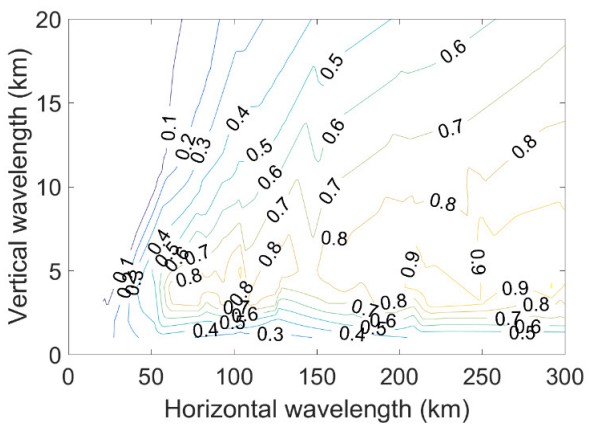

**Figure 9. Simulated sensitivity of the MATS instrument to gravity waves of varying horizontal and vertical wavelengths. Results are shown for waves with wave fronts perpendicular to orbit plane. The sensitivity is defined as the ratio between the amplitude of the**
**retrieved wave and the amplitude of the true wave field.**





**4.5 O₂ Atmospheric Band spectroscopy**

Following the tomographic retrieval of volume emission rates, the O₂ Atmospheric Band is analysed to reveal temperature and oxygen densities. Starting point for the temperature retrieval is the ratio, $R$, of the signals in the two Atmospheric Band channels. Figure 10 shows the (re-)distribution of the rotational transitions in the 0-0 vibrational band as a function of

5    temperature. With a lifetime of 12 s, the rotational distribution of $O_2$ ($b^1\Sigma$) is in thermodynamic equilibrium up to altitudes around 120 km, and thus representative for atmospheric temperature. The filter curves of the two MATS Atmospheric Band channels ("total" and "centre") have been chosen so that the ratio $R$ provides maximum sensitivity to temperature in the temperature range of interest.

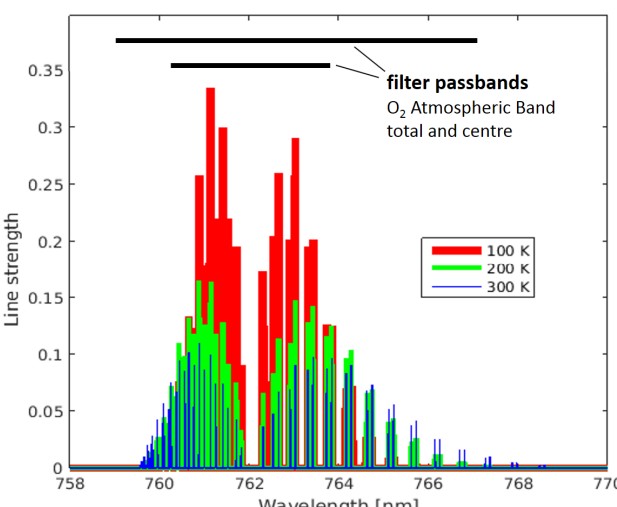

**Figure 10. Spectral distribution of the O₂ Atmospheric Band rotational transitions in the 0-0 vibrational band as a function of temperature. The filter passbands of the limb instrument's Atmospheric Band channels cover the total band and its central part, respectively.**

15    Using Gaussian error estimation and assuming the error from each channel is roughly equal, the random error in the retrieved temperature is

$$\Delta T = \sqrt{2}\frac{\Delta\epsilon}{\epsilon}\frac{dT}{dR}$$

(4)

where $\epsilon$ and $\Delta\epsilon$ are the retrieved volume emission and its RMS error, and dT/dR is the sensitivity of the temperature to the ratio between the signals in the two channels. In order to achieve a temperature precision of 2 K, a signal-to-noise ratio $\epsilon/\Delta\epsilon$

20    better than 500 is needed, with can typically be achieved between 90 and 100 km altitude. It should be noted that the RMS error depends not only on the instrument, but also the covariance matrices used in the retrieval. The tomographic retrieval will apply horizontal and vertical regularization to suppress noise in the retrieved field. Hence, based on the true performance of




the MATS instrument, further trade-off studies will be made between noise and spatial resolution in terms measurement integration times, pixel binning, and regularization.

The total volume emission rate of the Atmospheric Band provides direct information about the concentration of exited molecular oxygen $O_2$ ($b^1\Sigma$). This is also the basis for retrieving concentrations of ozone and atomic oxygen, which are
intimately linked to $O_2$ ($b^1\Sigma$) via dayglow photochemical reactions (Evans et al., 1988). As compared to the more complex daytime retrievals, the Atmospheric Band nightglow emission is largely only dependent on atomic oxygen, which allows for rather direct retrieval of atomic oxygen concentrations (Sheese et al., 2011).

## 4.6 NLC spectroscopy

The tomographic retrievals from the MATS UV channels provides the amount of scattered sunlight by NLC throughout the
3D retrieval grid. Additional NLC information will be available from the Atmospheric Band background channels, thus providing complementary spectral NLC data in the infrared. The ratio of NLC-scattered sunlight and solar irradiance provides the volume scattering coefficient $\beta$ in each retrieval pixel.

The amount of light scattered from ice particles depends largely on the ratio between the size of the particle and wavelength of the light. Rayleigh scattering applies to particles much smaller that the wavelength, with the scattering coefficient
approximately proportional to $\lambda^{-4}$. (Note that even in the Rayleigh limit the exponent is not exactly 4, as spectral dependence of the ice particles' index of refraction causes an additional wavelength-dependence.) For larger particles, interactions with the incoming sunlight get more complicated, and the scattering can be described as Mie scattering for spherical particles, or more complex numerical schemes for non-spherical particles (e.g. Mishchenko and Travis, 1998). Even in these more general cases, the wavelength dependence of the scattering in a limited spectral range can conveniently be described by a dependence $\lambda^{-\alpha}$,
where $\lambda$ is the wavelength, and $\alpha$ a size-dependent exponent, the so-called Ångström exponent (e.g. von Savigny et al., 2005). This Ångström description is frequently used in particle size retrievels, relating spectral measurements of particle scattering to theoretical descriptions of scattering as a function of particle size. For the two UV channels of the MATS limb instrument, the Ångström exponent is obtained as

$$\alpha = \frac{\log(\lambda_2) - \log(\lambda_1)}{\log(\beta_1) - \log(\beta_2)}$$

(5)

Once the Ångström exponent in each tomographic retrieval pixel is determined, this value can be compared to scattering simulations of different ice particle distributions. Figure 11 shows an example of a lookup table connecting particles sizes and Ångström exponent for the MATS UV wavelengths. It is important to note, however, that the information that can be retrieved about the NLC particle population is limited: The Ångström exponent provides a single piece of information, and can thus determine one parameter describing the size distribution, e.g. a mode radius. This makes it necessary to make assumptions
about additional parameters describing the particle population. Here we use the same assumptions that have been used in earlier retrieval studies e.g. for the AIM/CIPS or Odin/OSIRIS instrument. This includes spheroid ice particles with an axial ration of





0.5, and a normal distribution of particle sizes with a distribution width that varies with the mode radius (Lumpe et al., 2013; Hultgren and Gumbel, 2014). As Figure 11 shows, the resulting relationship between Ångström coefficient and particle sizes will generally be ambiguous for larger mode radii exceeding about 100 nm. A method to remove this ambiguity is to involve information from the infrared channels in the retrieval (Karlsson and Gumbel, 2004). Physically, NLC particle populations are

expected to have mode radii below 100 nm. Once particle size information has been inferred in the form of a mode radius, absolute scattering coefficient and size information can be combined to also retrieve the local ice content (ice mass density) of the cloud.

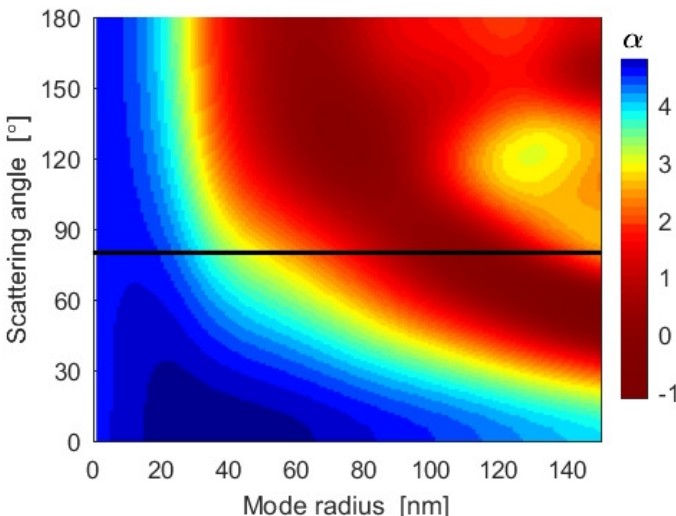

**Figure 11. A lookup table for the NLC particle size analysis, generated from T-matrix simulations. Shown is the Ångström exponent as a function of mode radius and scattering angle. The black line indicates a typical scattering angle for MATS NLC observations.**

Through Gaussian error propagation the error in the Ångström parameter in a retrieval pixel can be estimated through

$$\Delta\alpha = \frac{\sqrt{2}}{\log(\lambda_1) - \log(\lambda_2)} \frac{\Delta\beta}{\beta}$$

(6)

where $\Delta\beta$ is the uncertainty of the retrieved volume scattering coefficient, assuming it is roughly equal for the two wavelengths. From this, the error in the mode radius and ice mass density can be estimated through scattering simulations. In order to achieve a precision of 0.25 in the Ångström exponent, a signal-to-noise ratio $\beta/\Delta\beta$ better than 50 is needed.

## 5 Operational Planning

During routine operations, MATS will spend most of the time taking images using nominal science modes defined for the
mission. Based on season and time of day, different atmospheric phenomena are to be observed and, hence, different imaging



channels will operate. Beyond the nominal science modes, certain measurements must be performed in order to characterize the instrument in orbit. These calibration modes will be carried out with certain intervals, and involve changes in control settings for both the payload and the platform. During normal operations a set of commands will be uploaded once per week. For commands that are required to be executed at a particular point in space, the timing of the commands will use predicted
satellite orbits, based on orbit data ("two-line elements") gathered up to two weeks in advance.

## 5.1 Science modes

When defining operational measurement modes, an important constraint is the total data volume as defined by the satellite's downlink capacity. Hence, compromises must be made concerning image resolution, image compression, sampling interval, geographical coverage, and number of channels in operation. "NLC modes" will be active during the NLC season, i.e. summer
months in either hemisphere. During that period, the UV limb channels are given priority to operate at high resolution at summer latitudes poleward of 45°. The resolution of the IR limb channels is kept moderate in order to keep data volumes down. "IR modes" will be active outside the NLC season and involve no data collection in the UV channels, which allows the IR channels to operate at higher resolution. The IR nadir camera will always be operated during nighttime, producing a rather small data volume. Tables 5 and 6 summarize basic instrument settings in NLC and IR modes. The resulting measurements
are the basis for the retrieval of the primary data product listed in Table 1.

In addition to the instrument settings in Tables 5 and 6, the platform attitude can be adapted in accordance with specific measurement objectives. A standard viewing geometry is to take subsequent limb images centred around the satellite orbit plane. At the highest latitudes, this provides aligned images as input tomographic retrieval. However, at lower latitudes the Earth rotation leads to a continuous shifting of the atmospheric scene with respect to the orbit plane. Mats provides the option
to compensate for this in terms of a continuous yaw movement of the satellite. This movement lets the limb field of view follow the Earth rotation, thus keeping a targeted section of the atmosphere aligned between subsequent limb images.



| NLC Mode (May 1 - September 10, November 1 - March 10) | | | | | | | |
|---|---|---|---|---|---|---|---|
| Channel | UV1 | UV2 | IR1 | IR2 | IR3 | IR4 | Nadir |
| Horizontal pixel-size (km) | 5 | 5 | 10 | 10 | 50 | 50 | 10×10 |
| Vertical pixel size (km) | 0.2 | 0.2 | 0.4 | 0.4 | 0.8 | 0.8 | N/A |
| Number of across-track pixels | 50 | 50 | 25 | 25 | 5 | 5 | 18.5 |
| Number of vertical pixels | 200 | 200 | 138 | 138 | 69 | 69 | N/A |
| Readout interval (s) | 3 | 3 | 5 | 5 | 5 | 5 | ~1,4 |
| Integration time (s) | <3 | <3 | <5 | <5 | <5 | <5 | 1 |
| JPEG quality (%) | 90 | 90 | 90 | 90 | 90 | 90 | N/A |
| Data per image (kB) | 4.31 | 4.31 | 1.49 | 1.49 | 0.15 | 0.15 | 0.08 |
| Active fraction of orbit (%) | 30 | 30 | 100 | 100 | 100 | 100 | 30 |
| Images per orbit | 540 | 540 | 1080 | 1080 | 1080 | 1080 | 1157 |
| Data per orbit (kB) | 2325 | 2325 | 1604 | 1604 | 160 | 160 | 96 |

**Table 5. Image readout in the six limb channels and the nadir channel during "NLC mode". Information about pixels refers to image pixels that are created from on-chip binning of the individual CCD pixels. The total image data amounts to 8275 kB per orbit.**

| IR Mode (March 11 - April 30, September 11 - October 31) | | | | | | | |
|---|---|---|---|---|---|---|---|
| Channel | UV1 | UV2 | IR1 | IR2 | IR3 | IR4 | Nadir |
| Horizontal pixel-size (km) | - | - | 5 | 5 | 50 | 50 | 10×10 |
| Vertical pixel size (km) | - | - | 0.4 | 0.4 | 0.8 | 0.8 | N/A |
| Number of across-track pixels | - | - | 50 | 50 | 5 | 5 | 18.5 |
| Number of vertical pixels | - | - | 138 | 138 | 69 | 69 | N/A |
| Readout interval (s) | - | - | 5 | 5 | 5 | 5 | ~1,4 |
| Integration time (s) | - | - | <5 | <5 | <5 | <5 | 1 |
| JPEG quality (%) | - | - | 94 | 94 | 94 | 94 | N/A |
| Data per image (kB) | - | - | 3.68 | 3.68 | 0.18 | 0.18 | 0.08 |
| Active fraction of orbit (%) | 0 | 0 | 100 | 100 | 100 | 100 | 10 |
| Images per orbit | 0 | 0 | 1080 | 1080 | 1080 | 1080 | 386 |
| Data per orbit (kB) | 0 | 0 | 3972 | 3972 | 199 | 199 | 32 |

**Table 6. Image readout in the six limb channels and the nadir channel during "IR mode". Information about pixels refers to image pixels that are created from on-chip binning of the individual CCD pixels. The total image data amounts to 8373 kB per orbit.**



## 5.2 Calibration modes

In addition to the above science modes, a number of measurements are to be performed to characterize the instrument in orbit. Calibration measurements will be carried out with certain intervals, and involve changes in control settings for both the payload and the platform. Moreover, special manoeuvres can be performed to verify the integrity of the instrument after launch or at other occasions.

The driver for these calibration and special modes is that instrument properties may change in time, and need to be monitored. Table 7 lists main properties in this regard, including satellite operations and time scales over which changes can occur. These properties may in turn be dependent on operational parameters like solar position, instrument temperature etc., and should be monitored together as a function of those, if applicable. For these characterization activities, dedicated attitude operations have been defined for the platform, e.g. providing pointing towards the lower (Rayleigh-scattering) atmosphere, dark space, the moon or stars. During these measurements the CCDs may be operated with different integration times, with reduced pixel binning or full image readout.

| Property | Satellite operation | Timescale |
|---|---|---|
| Dark current | Pointing into darkness. | Weeks |
| Readout bias | Pointing into darkness. | Months |
| Bad pixels | Pointing into darkness. | Weeks |
| Noise level | Pointing into darkness. | Months |
| Polarization sensitivity | Role motion, pointing to various altitudes. | Years |
| Stray light | Limb scanning from brighter lower altitudes to darkness. | Years |
| Point spread function | Pointing at stars. | Years |
| Relative spectral calibration | Pointing at moon. Pointing to lower altitudes. | Years |
| Absolute calibration | Pointing to lower atmosphere. Pointing at moon. | Months |
| Instrument pointing | Pointing at stars. | Months |

**Table 7. Properties of the MATS limb instrument that need to be characterized during the mission.**



## 6 Outlook

In the late 1950s, Georg Witt laid the foundation for mesospheric research in Sweden. Studying NLC by ground-based photography, he applied stereoscopic analysis to infer three-dimensional structures of the clouds (Witt, 1962). 60 years later, the MATS satellite is about to study three-dimensional structures in the mesosphere by tomographic observations from space.

Georg Witt passed away in 2014, but he was still with us when MATS was proposed and selected earlier the same year. His scientific ideas will be with us when MATS flies.

At the time this paper is written, MATS is being prepared for launch in late 2019. Platform, instruments, and system have passed Critical Design Reviews, and are now going through assembly, integration and testing. Pre-flight calibration procedures have been developed and are being applied to characterize instrument properties like sensitivity, spectral and polarisation

dependence, dark current and other CCD characteristics. In parallel, procedures and software are being developed for data handling, tomographic and spectroscopic retrieval, and scientific analysis.

Basic measurement targets in the upper mesosphere and lower thermosphere are $O_2$ Atmospheric Band airglow and NLC. From these, the primary data products emission rate, temperature, atomic oxygen and ozone, as well as NLC brightness and particle size will be retrieved (Table 1). Based on these three-dimensional data products, an analysis in terms of gravity waves

and other dynamical structures will be conducted. This needs essentially two steps: firstly, (wave) structures need to be identified, which involves appropriate filtering against noise and small-scale fluctuations; secondly accessible wave parameters like horizontal and vertical wavelengths, wave orientation, and momentum flux will be addressed. The resulting wave climatology can be analysed e.g. in terms of wave spectra, as a function of latitude and season, and in relationship to dynamic conditions and drivers in other parts of the atmosphere. Co-analysis with other missions and meteorological data will be central

to these efforts. Modelling efforts will be decisive to combine different parts of these studies into the larger picture of atmospheric dynamics.

At all stages of the above analysis, collaboration with other research groups will be necessary and highly welcome. This concerns both the MATS dynamics objectives and mesospheric cloud objectives as defined in Section 1.1. The launch of the MATS satellite will be followed by an intense period of consolidating observational data and retrieval methods. Initial work

on the scientific analysis will then be conducted by a core team of collaborating research groups. This will soon be followed by general releases of data products on the different levels.

As for NLC studies, there will be a natural connection to the Odin satellite mission, providing a comprehensive OSIRIS NLC climatology of 17 years so far (Gumbel and Karlssson, 2011). Beyond climatology, as the orbits of MATS and Odin will be in close proximity, there are also perspectives towards more direct co-analysis on an orbit-to-orbit basis. Based on

overlapping orbits, true common-volume NLC studies are envisaged between MATS and AIM/CIPS. This follows the path already laid out by common volume studies between AIM and Odin (Benze et al., 2017; Broman et al., this issue). Joint studies by CIPS and MATS will make use of the complementary nature of the missions with the highly resolved horizontal data of CIPS and the three-dimensional tomographic data of MATS.




An example of a "whole atmosphere" perspective is a collaboration envisaged between MATS and several NASA missions, together providing the potential of gravity wave studies ranging from the troposphere to the thermosphere. Various methods for gravity wave analysis have been developed for AQUA/AIRS from the troposphere to mid-stratosphere (e.g., Gong et al, 2012; Hoffmann et al., 2012), for AIM/CIPS in the stratopause region (Randall et al., 2017), and for the Global-scale

Observations of the Limb and Disk (GOLD) in the lower thermosphere (Greer et al., 2018). As a complement to these missions, the MATS gravity wave analysis fills an important gap in the upper mesosphere. As described in the introduction, these perspectives coincide with an era of increasing model abilities to explicitly simulate of gravity waves from tropospheric sources to effects in the middle and upper atmosphere (Watanabe et al., 2015; Becker and Vadas, 2018). An important basis for linking MATS results to the dynamics of the troposphere and stratosphere will also be the co-analysis with meteorological datasets.

In particular, high-resolution data from the Integrated Forecasting System (IFS) of the European Centre for Medium-Range Weather Forecasts (ECMWF) have been shown to well reproduce gravity wave activity throughout the stratosphere (Dörnbrack et al., 2017; Ehard et al., 2018).

On a local basis, the three-dimensional MATS data will provide new opportunities for joint studies with ground-based instrumentation. Ground-based networks with focus on wave analysis like the ARISE project will be of particular interest

(Blanc et al., 2018). In the field of NLC and related mesospheric ice phenomena, local measurements include lidars and MST radars. In the field of airglow, local measurements include ground-based nightglow imaging and related spectroscopic temperature analysis. Wave studies in the MLT will also benefit from coincident MATS tomography and local time- and altitude-resolved measurements of wind (e.g., by meteor radar) or temperature (e.g. by resonance lidars). On an even more detailed level, co-analysis is possible between three-dimensional MATS data fields and coincident sounding rocket

experiments. It is important to note that many of these local studies also provide valuable possibilities to validate MATS measurements and analysis methods. Extending local co-analysis into the thermosphere and ionosphere, the objectives of MATS are closely related to scientific goals of the EISCAT_3D incoherent scatter radar system (Aikio et al., 2014) and similar radar networks. Three-dimensional data fields from both measurement systems can open for intriguing new studies on dynamical structures and coupling processes across the MLT.

**Author contribution**

JG, LM, OMC, DPM, JS, BK and GW have contributed to the conceptional development of the MATS science mission. LM and OMC have been project leaders for the MATS instrument development; JG, NK and DPM have contributed to the overall coordination of the project. OMC, JD, GG, JG, AH, JH, NI, MK, AL, SM, LM, DPM, GO, JR and JS have worked with the development of instruments, retrieval and scientific analysis. SC, SP, WP and AH have designed the limb telescope. JG, DPM,

NI and BK have contributed to the acquisition of financial support for the mission. JG, OMC, LM and NK have prepared the manuscript with contributions from the co-authors.



**Competing interests**

The authors declare that they have no competing interests.

**Acknowledgements**

We thank the teams at Omnisys Instruments, OHB-Sweden, and ÅAC Mictrotec. Their dedicated engineering and management
work has made the MATS payload and the InnoSat/MATS satellite platform possible. The scientific development of the
mission has benefitted greatly from discussions with many scientists. We are particularly grateful to Gerd Baumgarten, Erich
Becker, Adam Bourassa, Doug Degenstein, Patrick Eriksson, Manfred Ern, Patrick Espy, Craig Haley, Mark Hervig, Martin
Kaufmann, Yvan Orsolini, Kristell Pérot, Dave Rusch, Kaoru Sato, Mike Stevens, Mike Taylor, Christian von Savigny, and
Kaley Walker. We thank Andreas Fjeldstad, Markus Janghede, Franz Kanngieser, Tobias Kuremyr, Daniel Pettersson, Simon
Pfreundschuh, Tejaswi Seth, and Sarah Zayouna for their contributions to instrument development, instrument characterization
and retrieval development. We thank Susanne Benze, Lina Broman, Koen Hendrickx, Maartje Kuilman, and Marin Stanev for
their engagement in the MATS science questions. The MATS project is funded by the Swedish National Space Agency
(SNSA), and we especially acknowledge the commitment of their representative Ronnie Lindberg. Additional financial support
has been provided by the Erna and Victor Hasselblad Foundation.

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
