# Peer review of "The MATS Satellite Mission – Gravity Waves Studies by Mesospheric Airglow/Aerosol Tomography and Spectroscopy"

_Atmospheric Chemistry and Physics, 2018_

## Referee Comment (RC1) · Anonymous Referee #1 · 23 Dec 2018

This manuscript describes a new soon-to-be-launched Swedish satellite mission. This mission will combine UV (for NLC) and IR (for the O2 (0,0) airglow emission band) limb observations with IR nadir imaging, to study the dynamics of the upper atmosphere, specifically gravity waves. This paper gives an overview of the scientific background and objectives, then it describes the instrument design and several analysis techniques, before explaining the operating modes and summarizing the mission. It is somehow unusual to comment on a paper giving an overview of a project which has probably been extensively reviewed before getting funded by the Swedish National Space Agency. I assume that most of the possible issues have been investigated already, so my comments will be more about lacking information, even if I understand

that a simple paper cannot cover all the technical details involved in such a mission, and some personal remarks.

How can the second scientific question be answered since there is no thermospheric measurements above ∼96 km, no ionospheric measurements or modeling involved in MATS? Does it depends on external collaborations? I don't understand some of the instruments parameters. For example, Tables 5 and 6 gives 10x10 km for the nadir resolution. In section 3.4, it is said that the fov will be 200x50 km, and in section 3.2.4, that the CDD format is 512x2048 px, therefore the spatial resolution should be 100x100 m. Is there any binning involved? How much? Or will the images be cropped? The authors should give the speed of the satellite, it is important to understand the effects due to smearing. The readout times are close to the exposure times (tables 5 and 6), and up to 5s, which seem rather large for a satellite moving at probably several km/s. 4.2.2 briefly mentions readout smearing but doesn't give much information or references about desmearing. It also doesn't give any errors on the GW parameters measurements due to this effect. Table 1 shows the expected measurement precisions. 5-20K for the O2 nighttime temperature is very large given that this range will correspond to the temperature perturbations expected for the most impactful GWs. What will be the error on momentum flux calculation? The satellite will fly on a sun-synchronous, near-terminator, polar orbit, which means it will be complicated to look at tidal effects. Several corrections rely on climatologies or models. Is it planned to improved the corrections in the future? Figure 9 shows the sensitivity to GW horizontal and vertical wavelengths for the limb instrument. The best "region" is for 3<Lz<10 km and Lh>∼60km. These waves won't be observable by the nadir instrument because of the integration through the O2 layer. How do you plan to combine limb and nadir results? Furthermore, a large part of the important GWs (the ones transporting a lot of momentum) will be poorly characterized, especially GWs with Lh<60 km. What is the expected impact on the GW and MF climatology? I understand some of these questions/comments may be out of the scope of this paper, but it would be interesting to provide some answers or clarifications.

[Figure]

Minor edits p3 l.11: Ern et al., 2011 p4 l.16: first sentence sounds weird l.20: Forbes et al., 2009 l.23: Funke et al., 2010 l.30: ...satellite measurements... p5 l.4: could include Azeem et al., 2015 l.13: Gong et al., 2012 l.15: combining p6 l.3: by Song et al. l.25: remove "in" p7 l.4-6: looks like there is only 1 science question concerning NLC p8 l.6: Sheese et al., 2010 l.9: Mlynczack et al., 2001 l.11: patterns l.13: von Savigny et al., 2005 p9 l.13-14: change [] to () for the references l.21: Llewellyn p15 l.12: line stops too early, maybe pdf issue p16 l.17: analog-to-digital l.22: section 0??? p17 l.12: ...is better than... l.17: Figure 6 shows... p21 l.7: field of view p22 l.14: a priori... a posteriori l.31: 175 km p23 l.5: a priori l.8: change +- to normal sign p26 l.1: in terms of... l.3: excited l.9: provide l.21: retrievals l.31: ratio p28 l.19: MATS

---

## Referee Comment (RC2) · Anonymous Referee #2 · 29 Dec 2018

**Review of Gumbel et al., "The MATS Satellite Mission-Gravity wave studies by mesospheric airglow/aerosol tomography and spectroscopy"**

This paper is an overview of the new Swedish satellite mission, MATS, which will be launched in 2019. It is thoroughly written, covering scientific objectives, instrumentations, retrieval methods for tomography, etc. No major concerns were found. Only some minor issues. Publication in the ACP LPMR special issue is recommended after some clarifications being made.

Some references cannot be found in the reference list.

Comments:

1. Page 1, line 16, "gravity wave interactions", interactions with what? And many places throughout the manuscript.

2 Abstract, "Major scientific objectives...". This sentence is redundant with the first sentence in the abstract.

3. Page 2, line 1, "large scale circulation" -→"large scale circulations"

4. page 2, line 5, "upwell at the summer pole", the summer polar region might be a better expression. Same to " the winter pole".

5.  page 2, line 9, nomenclature, why "PMCs" are not used in this paper? Aren't PMCs often used by satellite measurements?

6. page 2, line 22, "is today understood", is there a typo?

7. page 2, line 23, "important mechanisms and interactions", what mechanisms and interactions? Please specify.

8. page 3, line 6, "MLT data...", gravity wave observations in the MLT?

9. page 3, line 14, "Also concerning mesospheric ice formation and NLC..", this sentence jumped out of nowhere in a gravity wave paragraph. How about a separate paragraph on the importance of NLC studies?

10.       page 5, line 9, "extending below 100 km" and "extending below 10 km", this could cause some confusion. "Below 100 km" means below 100 km altitude? How about "extending shorter than 100 km wavelength"?

11. page 5, line 21, "NLC" should use plural. "NLCs"

12. page 7, line 16, "Both O atmospheric band airglow and NLC feature...". It is very difficult to convert gravity waves in NLCs to temperature, geopotential or wind perturbations of gravity waves, that could be useful for models. Can the authors comment on this?

13. Figure 1. What are the red and purple lines in the two boxes?

14. page 8, line 11, "patters" or "patterns"?

15. page 9, lines 8-10, "nadir" actually means "from below the satellite.

16. page 9, the paragraph below Table 1. The orbit is near terminator (sunrise or sunset). Are you sure the nadir imager can see nightglow?

17. page 15, line 12, "Since the MATS...", this sentence belongs to the paragraph below.

18. page 16, line 22, "Section 0"?

19. page 17, line 17, "Figure.." which figure?

20. page 21, line 26, "MSIS-90", a newer version of MSIS is available: https://en.wikipedia.org/wiki/NRLMSISE-00
Why chose to use an older version of MSIS?

---

## Referee Comment (RC3) · Anonymous Referee #3 · 21 Jan 2019

This manuscript describes a new Swedish satellite mission to study gravity waves in the MLT region by tomographic techniques, applied to airglow emission spectroscopy and scattered sunlight from NLC. This mission provides the possibility of imaging gravity waves in 3-D on a global basis, and has the potential to provide much-needed input to global models which currently rely on parameterization methods. This paper is well-written and lays out all technical and scientific aspects of the mission. I recommend publication after making minor revisions. The most important issue that I found lacking was that of handling the issue of polarization. Polarized light will undergo different

Interactive
comment

effects depending upon the angle of incidence on metallic mirrors. In fact, on reflection circularly-polarization light will likely be produced. It is well-known that sunlight is highly polarized, so that any instrument that responds differently, based on the geometry, will be affected, and unless these effects are handled during calibration, would have significant effects on the received signal. Only once was the word polarization used. I am sure the authors are well aware of this effect, and am puzzled why this was not discussed in their paper. Other editing comments: Before line 15: relevant "to" (the last word is missing) Line 12 add the word "been" in between the words "have" and "retrieved" Line 15: "combing"-> "combining" Line 26: "limb imaging also opens". Better to say "limb imaging also provides" Line 10, new section: "patters-> "patterns" Page 26, last sentence" "ration"-> "ratio" The axial ratio used by Lumpe et al in the T-matrix formulation of scattering was assumed to be 2, not 0.5.: from Lumpe et al: "The particles are assumed to be oblate spheroids with an axial ratio of 2" "explicitly simulate of GW's -omit the word "of" Gravity waves have also been extracted from AIM SOFIE data (not in NLC), and are discussed in various references, e.g. the following: Gao, H., G. G. Shepherd, Y. Tang, L. Bu, Z. Wang (2017), Double-layer structure in polar mesospheric clouds observed from SOFIE/AIM, Ann. Geophys, 35, 295-309, doi:10.5194/angeo-35-295-2017.

---

## Author Comment (AC1) · 6 May 2019

**acp-2018-1162**
**Author Response to Referee Comments**

We thank the three referees for their very useful comments. Please find below our responses. These are followed by a revised version of our manuscript, with all changes marked.

**Referee #1  (acp-2018-1162-RC1)**

This manuscript describes a new soon-to-be-launched Swedish satellite mission. This mission will combine UV (for NLC) and IR (for the O2 (0,0) airglow emission band) limb observations with IR nadir imaging, to study the dynamics of the upper atmosphere, specifically gravity waves. This paper gives an overview of the scientific background and objectives, then it describes the instrument design and several analysis techniques, before explaining the operating modes and summarizing the mission.

It is somehow unusual to comment on a paper giving an overview of a project which has probably been extensively reviewed before getting funded by the Swedish National Space Agency. I assume that most of the possible issues have been investigated already, so my comments will be more about lacking information, even if I understand that a simple paper cannot cover all the technical details involved in such a mission, and some personal remarks.

How can the second scientific question be answered since there is no thermospheric measurements above ~96 km, no ionospheric measurements or modeling involved in MATS? Does it depends on external collaborations?

*Yes, the connection to the ionospheric questions relies on collaboration with other missions and databases. Similarly, the connections to the lower atmosphere (first science question) will rely on external such collaborations. This is stated in the paragraph following the science questions, and some examples of collaborations are given in Section 6 (Outlook).*

I don't understand some of the instruments parameters. For example, Tables 5 and 6 gives 10x10 km for the nadir resolution. In section 3.4, it is said that the fov will be 200x50 km, and in section 3.2.4, that the CDD format is 512x2048 px, therefore the spatial resolution should be 100x100 m. Is there any binning involved? How much? Or will the images be cropped?

*In all channels, information from the individual CCD pixels will be binned into larger image pixels (during usual scientific operation as listed in Table 5 and 6). This is required in order not to exceed the data rate available from the satellite downlink. The caption of the tables clarifies that information given in the tables refers to binned image pixels. For the nadir channel, binning into 10x10 km image pixels is performed. Sampling with larger resolution would not be meaningful because of the nadir image smearing (along track) due to the satellite motion.*

The authors should give the speed of the satellite, it is important to understand the effects due to smearing. The readout times are close to the exposure times (tables 5 and 6), and up to 5s, which seem rather large for a satellite moving at probably several km/s. 4.2.2 briefly mentions readout smearing but doesn't give much information or references about desmearing. It also doesn't give any errors on the GW parameters measurements due to this effect.

*The different kinds of smearing are described in the manuscript:*

*In section 3.2.4, we refer to readout smearing during the CCD readout (row-shifting). The handling of this CCD readout smearing is discussed further in section 4.2.2.*

*At the end of section 2.2 and in section 3.4, we refer to smearing of the nadir images due to the satellite motion ("motion blur"). We have now added information about the satellite speed (~7 km s$^{-1}$) in section 3.4. Methods for compensating for motion smearing exist in terms of synchronized CCD row shifting ("push broom technique"), but are not applicable to the MATS nadir imager. The only measure against motion blur applied in the MATS nadir channel is a restriction of the exposure time to max 1 s. Together with the readout resolution of the nadir images (10x10 km), this results in a spatial resolution of about 15 km along track. For the limb channels, on the other hand, retrieval simulations show that motion blur is not a significant limitation for the spatial resolution achievable by the tomography. The satellite moves up to 35 km during image exposure, but this is still much shorter than the e.g. the 700 km limb line-of-sight through a 10 km thick airglow layer.*

Table 1 shows the expected measurement precisions. 5-20 K for the O2 nighttime temperature is very large given that this range will correspond to the temperature perturbations expected for the most impactful GWs.

*Indeed, the spectral analysis of the temperature works best during daytime when the O2 dayglow provides the best signal-to-noise ratio for Atmospheric Band temperature retrieval and tomographic wave analysis. At nighttime, the precision of the nightglow O2 Atmospheric Band temperature retrieval is limited. When providing temperature data with the maximum tomographic resolution of 60x20x1 km, the precision of the temperature will not be better than 5-20 K (Table 1). The temperature precision can be improved by spatial averaging (either in terms of image pixel binning or in terms of post-flight data processing), which of course will diminish the spatial resolution. However this trade-off is handled, the possibilities to obtain small scale gravity wave data from the temperature field are rather limited during nighttime. We have now added text in section 4.5 stating this more clearly. Nonetheless, good nighttime gravity wave data is expected from other retrieved data fields (nightglow volume emission rate, atomic oxygen).*

What will be the error on momentum flux calculation?

*The ability to derive momentum flux depends on observation condition and the spectral sensitivity of the wave retrieval to different horizontal and vertical wavelength. A detailed discussion of the possibilities and limitation of retrieving different wave properties is beyond the scope of the present manuscript and will be provided as part of future wave studies. In the current manuscript, we have moved our description of the wave retrieval to a new, separate section 4.7, and extended it with a basic account of the steps involved in the planned wave analysis.*

The satellite will fly on a sunsynchronous, near-terminator, polar orbit, which means it will be complicated to look at tidal effects.

*Yes, possibilities to infer tidal information from the MATS data will be very limited. At a given latitude, data will be obtained at two (rather) fixed local times only. This fact actually facilitates our efforts to obtain a consistent gravity wave climatology. Since the gravity wave*

*analysis is performed in a fixed phase of the (migrating) tide, variations due to tidal influences are minimized in our data set. A deeper analysis of relationships between gravity waves and tides will require co-analysis with external data sets.*

Several corrections rely on climatologies or models. Is it planned to improved the corrections in the future?

*With a "low-budget" mission like MATS it has not been possible to design a completely self-sustained system that can provide all information necessary from in-orbit measurements, on-board calibration etc. We thus need to rely on external data, climatologies, and simulations to obtain a complete set of information that is necessary for the data analysis.*

*An example is knowledge about the atmospheric density (or pressure) profile that is necessary e.g. for the retrieval of certain wave parameters or for the subtraction of molecular scattering background from the NLC measurements. There are ideas to use measurements of Rayleigh scattering in several channels to retrieve limited density information. However, such approaches are difficult to quantify before actual flight data become available, and we thus do not want to speculate in the manuscript about other possibilities than using climatologies or model atmospheres as a source for atmospheric density.*

Figure 9 shows the sensitivity to GW horizontal and vertical wavelengths for the limb instrument. The best "region" is for $3 < L_z < 10$ km and $L_h > \sim 60$ km. These waves won't be observable by the nadir instrument because of the integration through the $O_2$ layer. How do you plan to combine limb and nadir results?

*As we state in the manuscript, possibilities are limited to obtain useful data from the nadir channel and to combine these data with the limb analysis. The nadir observations concern $O_2$ nightglow and require sufficient darkness (large solar zenith angles) in order to provide useful data. The near-terminator orbit chosen for MATS is not optimized for these nadir observations and will provide sufficient darkness largely only during high-latitude winter conditions. The tomographic limb analysis, on the other hand, works best during daytime when $O_2$ dayglow provides the best signal-to-noise ratio for temperature retrieval and tomographic wave analysis. At nighttime, the precision of the nightglow $O_2$ Atmospheric Band temperature retrieval is limited (Table 1), requiring spatial averaging and thus prohibiting wave analysis down to the small horizontal scales accessible in daytime. (We have added more text in Section 4.5 to clarify this. See also our response above to the reviewer's question about the $O_2$ nighttime temperature) The possibilities for combined wave analysis from nadir and limb observations are therefore rather limited during this mission. In this sense, the addition of the nadir imager may be regarded as an interesting test of an observational concept rather than an essential part of this mission. Nonetheless, we do expect exciting case studies with detailed nadir observations of wave events in the $O_2$ Atmospheric Band nightglow, in combination with limb analysis of atmospheric background conditions in terms of temperature and odd oxygen fields.*

Furthermore, a large part of the important GWs (the ones transporting a lot of momentum) will be poorly characterized, especially GWs with $L_h < 60$ km. What is the expected impact on the GW and MF climatology? I understand some of these questions/comments may be out of the scope of this paper, but it would be interesting to provide some answers or clarifications.

*Figure 9 in the original manuscript (now Figure 11 in the revised manuscript) only shows the retrieval sensitivity for waves with wave fronts perpendicular to the orbit plane. As described in the text, because of the higher retrieval resolution across track, waves with wave fronts parallel to the orbit plane can be inferred with horizontal wavelengths down to 20 km. This asymmetry in the retrieval resolution between along track and across track is a complication for our data analysis. However, it means that in most cases, waves with horizontal wavelengths well below 60 km can be identified. This is true in particular since gravity waves tend to travel (anti)parallel to the mean wind, i.e. preferably in east-west direction, and thus with their wave fronts oriented parallel to the orbit plane (close to north-south for most of the orbit). In the extended Section 4.7, we briefly describe the need to specify a "sensitivity function" that defines the response of our retrievals in terms of a wave's wavelength and orientation as well as observation conditions. This is necessary for a proper analysis of gravity wave and momentum flux spectra. A deeper discussion is indeed beyond the scope of the current overview paper.*

Minor edits

*The minor edits listed below have been corrected. We apologize for leaving so many misspellings and stylistic errors in the manuscripts, and we thank the reviewer for a thorough work in identifying those.*

p3 l.11: Ern et al., 2011
p4 l.16: first sentence sounds weird
p4 l.20: Forbes et al., 2009
p4 l.23: Funke et al., 2010
p4 l.30: ...satellite measurements...
p5 l.4: could include Azeem et al., 2015
p5 l.13: Gong et al., 2012
p5 l.15: combining
p6 l.3: by Song et al.
p6 l.25: remove "in"
p7 l.4-6: looks like there is only 1 science question concerning NLC
p8 l.6: Sheese et al., 2010
p8 l.9: Mlynczack et al., 2001
p8 l.11: patterns
p8 l.13: von Savigny et al., 2005
p9 l.13-14: change [] to () for the references
p9 l.21: Llewellyn
p15 l.12: line stops too early, maybe pdf issue
p16 l.17: analog-to-digital
p16 l.22: section 0???
p17 l.12: ...is better than...
p17 l.17: Figure 6 shows...
p21 l.7: field of view
p22 l.14: a priori... a posteriori
p22 l.31: 175 km
p23 l.5: a priori
p23 l.8: change +- to normal sign
p26 l.1: in terms of...

p26 l.3: excited
p26 l.9: provide
p26 l.21: retrievals
p26 l.31: ratio
p28 l.19: MATS

**Referee #2** (acp-2018-1162-RC2)

This paper is an overview of the new Swedish satellite mission, MATS, which will be launched in 2019. It is thoroughly written, covering scientific objectives, instrumentations, retrieval methods for tomography, etc. No major concerns were found. Only some minor issues. Publication in the ACP LPMR special issue is recommended after some clarifications being made.

Some references cannot be found in the reference list.
*The reference has been checked and some references have been added.*

Comments:

1. Page 1, line 16, "gravity wave interactions", interactions with what? And many places throughout the manuscript.
*We usually refer to gravity wave interactions both with larger scale waves and with the mean flow. This has now been added explicitly in some (but not all) places.*

2 Abstract, "Major scientific objectives…". This sentence is redundant with the first sentence in the abstract.
*We have changed this. The first sentence is now a general statement, while the latter is about the wave interactions (see above).*

3. Page 2, line 1, "large scale circulation" -à"large scale circulations"
*Corrected.*

4. page 2, line 5, "upwell at the summer pole", the summer polar region might be a better expression. Same to " the winter pole".
*Corrected.*

5. page 2, line 9, nomenclature, why "PMCs" are not used in this paper? Aren't PMCs often used by satellite measurements?
*Some scientists indeed talk about PMC when observing from satellites, and NLC when observing in other ways. But it is the same thing, and there is no general rule. The MATS project very much builds on the heritage of Georg Witt, and in his spirit we stick to the nomenclature NLC.*

6. page 2, line 22, "is today understood", is there a typo?
*We have clarified the statement by moving "today" to an earlier part of the sentence.*

7. page 2, line 23, "important mechanisms and interactions", what mechanisms and interactions? Please specify.

*This is now specified in the subsequent sentences about wave sources, dissipation and forcing, with momentum flux as the decisive quantity.*

8. page 3, line 6, "MLT data…", gravity wave observations in the MLT?
*This has been adopted.*

9. page 3, line 14, "Also concerning mesospheric ice formation and NLC..", this sentence jumped out of nowhere in a gravity wave paragraph. How about a separate paragraph on the importance of NLC studies?
*This is a good suggestion. We have done so.*

10. page 5, line 9, "extending below 100 km" and "extending below 10 km", this could cause some confusion. "Below 100 km" means below 100 km altitude? How about "extending shorter than 100 km wavelength"?
*This is done.*

11. page 5, line 21, "NLC" should use plural. "NLCs"
*Corrected.*

12. page 7, line 16, "Both O atmospheric band airglow and NLC feature…". It is very difficult to convert gravity waves in NLCs to temperature, geopotential or wind perturbations of gravity waves, that could be useful for models. Can the authors comment on this?
*We state "Both O2 Atmospheric Band airglow and NLC feature horizontal and vertical structures that are a direct response to gravity wave activity." This does not mean that vertical structures in NLC can be used to retrieve vertical gravity wave properties. On the contrary, the wave analysis of NLC data will be restricted to horizontal wave structures. This is described later in this Section 2.2.*

13. Figure 1. What are the red and purple lines in the two boxes?
*The lines represent limb lines of sight through a given volume for an Atmospheric Band channel and an NLC channel, respectively. This is now explained more clearly in the figure caption.*

14. page 8, line 11, "patters" or "patterns"?
*Corrected.*

15. page 9, lines 8-10, "nadir" actually means "from below the satellite.
*Yes. Still, it makes sense to state this explicitly in this introductory sentence about the nadir imager.*

16. page 9, the paragraph below Table 1. The orbit is near terminator (sunrise or sunset). Are you sure the nadir imager can see nightglow?
*Indeed, sufficiently dark nadir measurement conditions will be restricted to the winter season at mid and high latitudes, i.e. a minor part of the total mission. This is now more clearly stated in this paragraph.*

17. page 15, line 12, "Since the MATS…", this sentence belongs to the paragraph below.
*Corrected.*

18. page 16, line 22, "Section 0"?
*Corrected.*

19. page 17, line 17, "Figure.." which figure?
*Corrected.*

20. page 21, line 26, "MSIS-90", a newer version of MSIS is available: https://en.wikipedia.org/wiki/NRLMSISE-00. Why chose to use an older version of MSIS? *We have now updated this. The use of MSIS-90 was a heritage from the NLC analysis of the Odin satellite mission.*

**Referee #3**  (acp-2018-1162-RC3)

This manuscript describes a new Swedish satellite mission to study gravity waves in the MLT region by tomographic techniques, applied to airglow emission spectroscopy and scattered sunlight from NLC. This mission provides the possibility of imaging gravity waves in 3-D on a global basis, and has the potential to provide much-needed input to global models which currently rely on parameterization methods. This paper is well written and lays out all technical and scientific aspects of the mission. I recommend publication after making minor revisions.

The most important issue that I found lacking was that of handling the issue of polarization. Polarized light will undergo different effects depending upon the angle of incidence on metallic mirrors. In fact, on reflection circularly-polarization light will likely be produced. It is well-known that sunlight is highly polarized, so that any instrument that responds differently, based on the geometry, will be affected, and unless these effects are handled during calibration, would have significant effects on the received signal. Only once was the word polarization used. I am sure the authors are well aware of this effect, and am puzzled why this was not discussed in their paper.

*As the referee states, polarisation will have an impact on the received signal. Some text has been added to highlight this in section 4.3. Based on pre-flight analysis, a main cause of polarisation response in the MATS instrument are the beamsplitters mounted at 45 degree to the incoming beam. At this moment of writing, the laboratory analysis of the instrument's polarization sensitivity is still ongoing as part of the calibration and characterisation procedures.*

*Different contributions to the total detected signal in orbit (airglow emission, NLC scattering, molecular scattering, including scattering of both direct sunlight and upwelling radiation) have different polarisation properties. As these contributions cannot be separated from each other based on the in-orbit measurements alone, assumptions need to be made during the radiometric calibration of the images. For the different measurement channels the following assumptions will be made:*

*1. UV: Single scattering will be assumed both for the light scattered of NLCs and the background Rayleigh scattering. This can be assumed as upwelling radiation from the lower atmosphere in the UV to a large extent is absorbed by the ozone layer.*

*2. IR: In the O2 Atmospheric band region the scattering geometry is different. For the airglow emission unpolarised radiation can be assumed but for the scattered background radiation it is more complex. We have carried out an analysis using the SASKTRAN radiative transfer simulator to estimate the variability in polarisation of the scattered background. These simulations indicate that the background polarisation ratio will vary from between 0.1 to 0.2 depending on the albedo of the lower atmosphere. These differences are negligible at the peak altitudes of the A-band emission, but will have a significant impact on the temperature retrieval (>5 K) at the lower and upper boundaries. However, it is expected that this uncertainty can be reduced by utilizing data from the albedo photometers to quantify the contribution of upwelling radiation to the scattered background radiation.*

Other editing comments:

Before line 15: relevant "to" (the last word is missing)
*Corrected.*

Line 12 add the word "been" in between the words "have" and "retrieved"
*Corrected.*

Line 15: "combing"-> "combining"
*Corrected.*

Line 26: "limb imaging also opens". Better to say "limb imaging also provides"
*Corrected.*

Line 10, new section: "patters-> "patterns"
*Corrected.*

Page 26, last sentence" "ration"-> "ratio" The axial ratio used by Lumpe et al in the T-matrix formulation of scattering was assumed to be 2, not 0.5.: from Lumpe et al: "The particles are assumed to be oblate spheroids with an axial ratio of 2"
*Yes, this is correct. We have been aware that the CIPS retrieval assumes oblate spheroids, but we have misinterpreted the axial ratio and referred to its inverse in our manuscript. This has now been corrected.*

"explicitly simulate of GW's -omit the word "of"
*Corrected.*

Gravity waves have also been extracted from AIM SOFIE data (not in NLC), and are discussed in various references, e.g. the following: Gao, H., G. G. Shepherd, Y. Tang, L. Bu, Z. Wang (2017), Double-layer structure in polar mesospheric clouds observed from SOFIE/AIM, Ann. Geophys, 35, 295-309, doi:10.5194/angeo-35-295-2017.
*We missed out on this gravity wave data product from SOFIE. We have now added it to the manuscript.*

**Other:**

*Sections 4.2 (calibration) and 4.3 (background removal) have undergone some rewriting. This comprises a discussion of polarization (in response to Referee #3) but also other improvements.*

*The description of gravity wave retrievals has been extended and moved from section 4.4.3 to the new section 4.7. This provides a more logical structure to the overall description of the data analysis (first tomography, then spectroscopy, then wave retrieval), and also follows the reviewers' questions about the planned wave analysis.*

*Some new references have been added:*

[revised manuscript text omitted]

(a)

M1
M2
M3

(b)

BS UV IR    filterB IR    BS IR4              FM IR

                            filterN              filterN
          FM UV    BS IR5    IR2    BS IR2    IR4
BS UV

filterB    filterB                    CCD              CCD
UV1        UV2              channel IR2    channel IR4
                            ABandTotal        BG-Long

filterN    filterN    filterN          filterN
UV1        UV2        IR3              IR1
CCD        CCD        CCD              CCD
channel UV1 channel UV2 channel IR3    channel IR1
UV-Short    UV-Long    BG-Short        ABandCenter

[revised manuscript text omitted]